# Evaluating explainability techniques on discrete-time graph neural networks

**Manuel Dileo**                                             *manuel.dileo@unimi.it*
*Department of Computer Science*
*University of Milan, Milan, Italy*

**Matteo Zignani**                                          *matteo.zignani@unimi.it*
*Department of Computer Science*
*University of Milan, Milan, Italy*

**Sabrina Gaito**                                            *sabrina.gaito@unimi.it*
*Department of Computer Science*
*University of Milan, Milan, Italy*

**Reviewed on OpenReview:** *https://openreview.net/forum?id=JzmXoOrfry*

## Abstract

Discrete-time temporal Graph Neural Networks (GNNs) are powerful tools for modeling evolving graph-structured data and are widely used in decision-making processes across domains such as social network analysis, financial systems, and collaboration networks. Explaining the predictions of these models is an important research area due to the critical role their decisions play in building trust in social or financial systems. However, the explainability of Temporal Graph Neural Networks remains a challenging and relatively unexplored field. Hence, in this work, we propose a novel framework to evaluate explainability techniques tailored for discrete-time temporal GNNs. Our framework introduces new training and evaluation settings that capture the evolving nature of temporal data, defines metrics to assess the temporal aspects of explanations, and establishes baselines and models specific to discrete-time temporal networks. Through extensive experiments, we outline the best explainability techniques for discrete-time GNNs in terms of fidelity, efficiency, and human-readability trade-offs. By addressing the unique challenges of temporal graph data, our framework sets the stage for future advancements in explaining discrete-time GNNs.

## 1 Introduction

Temporal Graph Neural Networks (TGNNs) have emerged as a crucial tool for modeling evolving graph-structured data (Longa et al., 2023). These models are widely applied in various domains, including social network analysis (Dileo et al., 2023), financial systems (Pareja et al., 2020; Dileo & Zignani, 2024), and collaboration networks (You et al., 2022), where understanding dynamic relationships is critical. Discrete-time TGNNs, in particular, represent temporal data as a series of discrete snapshots, providing a flexible and scalable approach for capturing temporal dependencies. Despite their effectiveness, the black-box nature of TGNNs raises significant challenges in interpreting their predictions, which is essential for building trust and ensuring transparency in sensitive applications such as financial fraud detection or social behavior analysis.

Explainability techniques aim to address this challenge by providing insights into the decision-making processes of machine learning models. While considerable progress has been made in explaining static GNNs (Longa et al., 2025), the unique temporal and dynamic nature of TGNNs introduces additional complexities that remain largely unexplored. Temporal data often involves evolving patterns (Paranjape et al., 2017), recurrent interactions (Gastinger et al., 2024), and distribution shifts (Zhang et al., 2022), which must be

carefully considered when generating and evaluating explanations. Although some approaches for explaining TGNNs have been proposed in the literature (Chen & Ying, 2023; Xia et al., 2023), none of these methods are specifically tailored for discrete-time temporal networks. Furthermore, their evaluation strategies and baselines rely heavily on the conventional setting developed for explaining static GNNs (Amara et al., 2022).

Hence, in this work, we propose a novel framework to evaluate explainability techniques in this context. Our framework addresses the temporal characteristics of the data and models by introducing new training and evaluation settings, temporal-specific metrics, and tailored baselines. Through our experiments, we provide practical guidance on the trade-offs among fidelity, efficiency, and human-readability of the current state-of-the-art techniques, including graph and gradient-based techniques.

**Contributions**  We summarize our main contributions as follows: *i)* **Training and evaluation setting:** We propose a new setting that takes into account the evolving nature of the data when generating and evaluating explanations for discrete-time temporal networks; *ii)* **Evaluation metrics:** We extend and complement the set of evaluation metrics to focus on the temporal aspects of the provided explanations, such as the recurrence of the events. *iii)* **New baselines for explaining TGNNs:** We propose baselines and models tailored for discrete-time GNNs, by leveraging simple ideas like considering only the most recent events or by extending well-known explainability techniques such as GNNExplainer; *iv)* **Practical guidance:** We conduct an extensive evaluation to outline the current best explainability techniques for discrete-time GNNs in terms of fidelity, performance, and readability trade-offs.

**Scope**  The objective of our work is to advance research in explaining any discrete-time GNN model. Hence, we mainly focus on evaluating instance-level post-hoc explanation methods since most of the temporal graph models in the literature are not designed with specific consideration of explanations. The full definitions and taxonomy of explainability techniques are defined in Amara et al. (2022).

## 2  Background

**Temporal Graph Neural Networks**  Temporal graph neural networks (TGNNs) are deep learning models for extracting, learning, and predicting from evolving networks. The most relevant graph neural network architectures for temporal graphs are covered in a recent survey (Longa et al., 2023). Current TGNNs approaches can be divided based on how they model temporal networks. In continuous-time dynamic graph models, temporal networks are treated as a stream of ordered timestamped edges. In contrast, in discrete-time or snapshot-based dynamic graph models, temporal networks are a collection of ordered timestamped static graphs - snapshots - with a certain time granularity. These techniques are designed to handle two different kinds of temporal networks, and their training and evaluation strategies lead to two different settings (Huang et al., 2023b). For instance, event-based dynamic graph models can be used for streaming data, such as sensor data, or to predict anomalies with very high-resolution temporal precision (Reha et al., 2023). In contrast, snapshot-based models can be leveraged where data are collected with regular time intervals or in complex systems where nodes act very similarly in close timestamps, such as online social networks, collaboration, or financial networks (Pareja et al., 2020). In this paper, we focus on the latter models. Specifically, we provide a new setting, metrics, baselines, and models for evaluating explanations provided by discrete-time TGNNs models on future link prediction, the most well-known and studied prediction task on temporal networks (Huang et al., 2023b).

**Graph Explainers**  Explainability methods for Graph Neural Networks are generally divided into two categories: non-generative and generative methods. For a given input instance and its prediction, non-generative methods often employ approaches such as gradients (Baldassarre & Azizpour, 2019), perturbations (Huang et al., 2023a), mask optimization (Ying et al., 2019), surrogate models (Vu & Thai, 2020), subgraph matching (Wu et al., 2023), and Monte Carlo Tree Search (MCTS)(Yuan et al., 2021) to identify the explanation subgraph. These methods incrementally optimize the explanation during the explanation stage, which can result in a longer inference time. In contrast, generative methods involve training a generative model on the entire dataset to capture the distribution of the underlying explanatory subgraphs (e.g. Luo et al. (2020); Miao et al. (2022)), enabling a comprehensive understanding of model behavior across the dataset.

Compared to static GNNs, the explainability of Temporal Graph Neural Networks remains a challenging and relatively unexplored area. TGNNExplainer (Xia et al., 2023) is the first explainer specifically designed for temporal GNNs, utilizing the MCTS algorithm to identify a combination of explanatory events. TempME (Chen & Ying, 2023), instead, is derived from the information bottleneck principle and extracts the most interaction-related motifs while minimizing the amount of contained information to preserve the sparsity and succinctness of the explanation. However, these two works are tailored for continuous-time models. Recently, two other works proposed explainability techniques tailored for time series on graphs, i.e. spatiotemporal forecasting (He et al., 2022; Guerra et al., 2024). To the best of our knowledge, there are no methods tailored for the explainability of discrete-time graph neural networks. Hence, the focus of this work is to provide a comprehensive framework to advance this research area.

## 3   Problem setup

A temporal network can be modeled as a graph $\mathcal{G} = (V, E, T)$, where $V$ is the set of nodes, $E \subseteq V \times V = \{(u, v, t) | u, v \in V, t \in T\}$ is the set of interaction events - timestamped edges - and $T$ is the set of timestamps. A discrete-time temporal network $\mathcal{G} = \{G_t\}_{t=1}^{|T|}$ can be represented as a sequence of graph snapshots, where each snapshot is a static graph $G_t = (V_t, E_t)$ associated to a specific timestamp $t \in T$, which contains the interactions occurred at time $t$ and the nodes involved in them. We also denote with $\mathcal{G}_{[t_s, t_e]}$ the sequence of graph snapshot $G_t$ s.t. $t_s \leq t \leq t_e, t_s, t, t_e \in T$.

A discrete-time Graph Neural Network (GNN) takes as input a discrete-time temporal network $\mathcal{G}$ and learns a time-aware embedding for each node $v \in V$ up to a certain time $t$. For simplicity, we refer to discrete-time GNNs as temporal graph neural networks (TGNNs). TGNNs' capability for representing learning on temporal graphs is typically evaluated using the future link prediction task (Pareja et al., 2020; You et al., 2022; Huang et al., 2023b), i.e. predicting future interactions based on historical events. Hence, in this work, we focus on explaining the link prediction behavior of TGNNs.

Given a pre-trained TGNN $f$, our objective is to obtain an explanation model. Specifically, given a target interaction event $e$, an explainer for TGNNs aims at identifying a subset of important historical events that trigger the future interaction prediction made by the base model $f$. An "explanation" in the domain of GNNs is a mask or a subgraph of the initial graph, i.e., a set of weighted nodes, edges, and possibly node features (Amara et al., 2022). The weights on these graph entities relate to their inherent importance for explaining the model outcomes. The explainer model usually performs a feature attribution operation which associates each feature of the computation graph for the target event with a weight or relevance score for the link predictor. The computation graph of a target event $e = (u, v, t)$, denoted as $\mathcal{G}(e)$, might be the augmented graph of all the events up to time $t$, a subview of the graph based on a temporal window, or a subgraph around nodes $u$ and $v$, since some methods only look at a $k$-hop neighborhood to make predictions. The set of events in $\mathcal{G}(e)$ are called *candidate events*. In this study, we focus on the contribution of historical interaction events, namely the edges. To explain each target event $e$, all the methods compared in this paper generate a mask $\mathbf{M}_E \in \mathbb{R}^{|V| \times |V|}$, whose elements denote the importance score of a candidate event to the prediction of the target event. Finally, an explanation corresponds to a mask $\mathbf{M}_E$ on the edges that operate on the computational graph to form a subgraph $G_{exp}(e)$, where only edges with an importance score greater than zero are included.

## 4   Evaluating XAI on discrete-time GNNs

**Aspect 1: setting**

We propose a new setting that takes into account the evolving nature of the data when generating and evaluating explanations for discrete-time temporal networks.

**Problem 1:** Traditionally, discrete-time graph neural networks are trained in the deployed setting (Pareja et al., 2020). In this setting, no information from the test set is passed to the model, and the node embeddings from the last training snapshot are used for predictions in all test snapshots. Using the deployed setting

leads to generating explanations for future events based on the train set graph only, without considering all the events from the end of the train set until the target event, providing explanatory subgraphs that may be poorly informative.

**Solution:**  To avoid this behavior, we modify the traditional training and inference steps for discrete-time GNNs. First, the models are trained using the live-update setting (You et al., 2022) on the train set. This recently proposed strategy boosts the performance of discrete-time GNNs, where the dataset is split chronologically into snapshots and the model weights are constantly updated to newly observed snapshots while predicting the next one. This experimental setting represents an instance of prequential setting (Dawid, 1992), more suitable for training and evaluating machine learning algorithms for data streams. Then, we allow models to incorporate newly observed information in the test set to make future predictions. Specifically, no information from the test set is used to train the models but events after the train set are leveraged at inference time to update the node embeddings until the snapshot before the target event. In this way, explanations are generated based also on the newly observed events. In a concurrent work (Huang et al., 2024), the authors develop a very similar strategy, called UTG, to allow a comparison between discrete and continuous-time dynamic graph models. Our strategy can be considered as a slight variation of UTG where the training phase is performed using the live-update strategy.

**Problem 2:**  In previous works (Xia et al., 2023; Chen & Ying, 2023), the quality of the explanations for TGNNs is computed by selecting $k$ future target events to explain globally, i.e. uniformly at random over the entire test set, obtaining an evaluation of the overall quality of the explanations for future events. However, they do not evaluate the quality of the explanations over time, which may tell several useful information about the behavior of explainability techniques along the timestamps in the test set, such as the difficulty of providing good explanations for timestamps distant from the training set, or the identification of test time interval where they may be a distribution shift respect to the train set. In general, prior works assume that the quality of the explanations on future events may be similar for the entire lifetime of the dataset, which may be a rough approximation.

**Solution:**  To overcome this limitation, we select $k$ future target events to explain locally, i.e. for each test set snapshot. So, the evaluation metrics can be computed for each snapshot, and the performance trend over the test set can be easily shown.

We summarize our new strategy for generating and evaluating explanations for discrete-time temporal networks in Algorithm 1.

### Aspect 2: metrics

Previous works evaluate explanations for TGNNs using *fidelity sufficiency* (Xia et al., 2023; Chen & Ying, 2023), a very popular explainability metric for GNNs (Amara et al., 2022). Here, we extend fidelity to take into account the temporal aspect of the solutions. Furthermore, TempME introduces *cohesiveness* to evaluate explanations from a human-intelligible viewpoint, based on the temporal proximity of the adjacent events. Here, we propose to complement cohesiveness with new metrics that take into account other important structural aspects of temporal networks, such as the recurrence of the events.

**Fidelity sufficiency.**  In the context of future link prediction, for each event $e$, fidelity sufficiency measures the agreement on the edge existence between the prediction by the GNN when fed with the entire computational graph $\mathcal{G}(e)$ and the prediction when fed with the explanation of event $\mathcal{G}_{exp}(e)$. Given $K$ the set of target events, fidelity is defined as:

$$Fid(K) = 1 - \frac{1}{|K|} \sum_{e \in K} \mathbb{1}(f(\mathcal{G}_{exp}(i)) = f(\mathcal{G}(i)))$$

where $f(\mathcal{G}_{exp}(e))$ indicates the output of the GNN at the inference step when fed with the explanation of event $e \in K$. Usually, we want the explanation to be sufficient; i.e. it leads to the initial prediction of the

---

**Algorithm 1** Evaluating discrete-time GNNs explanations

---

**Input**: TGNN model, XAI technique, $D$ temporal annotated dataset
**Parameter**: $k$ number of events to explain
**Output**: Explanations and metrics

1: Split $D$ in train, validation and test set, chronologically
2: Let $t'$ be the first timestamp after the train set
3: Train GNN on $D$'s train set using the live-update setting. Optimize parameters using $D$'s validation set.

4: **for all** timestamps $t$ in D's test set **do**
5:     Let $G_t$ be the current snapshot.
6:     Let $G_{[t',t-1]}$ be the graph with all the events from the end of the train set until the previous snapshot.

7:     **for all** snapshots $G_{\hat{t}}$ in $G_{[t',t-1]}$ **do**
8:         Let $X_{\hat{t}}$ be the node feature matrix
9:         Let $E_{\hat{t}}$ be the edges of the snapshot
10:        Let $H_{\hat{t}-1}$ be the previous node embeddings
11:        $H_{\hat{t}} = \text{TGNN}(X_{\hat{t}}, E_{\hat{t}}, H_{\hat{t}-1})$
12:    **end for**
13:    Sample $\frac{k}{2}$ positive and $\frac{k}{2}$ negative edges as target events from $G_t$.
14:    Explain the target events using the XAI technique on the trained TGNN considering edges in $G_{[t',t-1]}$ as candidate events.
15:    Evaluate the explanations of the current snapshot.
16: **end for**
17: **return** explanations and metrics

---

model explanation. Hence, an explainability model that provides sufficient explanation has a fidelity close to zero. We later consider and report $(1 - Fid)$ in our experiments.

Given the evaluation procedure defined in Algorithm 1, in our scenario we have a set $K$ of target events for each test snapshot. Hence, we can extend this metric to obtain the *temporal fidelity sufficiency*, defined as:

$$TFid = \frac{1}{T_{test}} \sum_{j=1}^{T_{test}} (1 - Fid(K_j))$$

where $K_j$ denotes the set of target events related to test snapshot $j$, and $T_{test}$ is the number of snapshots in the test set.

**Cohesiveness.** Given a target event $e$, its cohesiveness is defined as:

$$Coh(e) = \frac{1}{|\mathcal{G}_{exp}(e)|^2 - |\mathcal{G}_{exp}(e)|} \sum_{i \in \mathcal{G}_{exp}(e)} \sum_{j \in \mathcal{G}_{exp}(e); i \neq j} \cos\left(\frac{|t_i - t_j|}{\Delta T}\right) \mathbb{1}(i \sim j),$$

where $\Delta T$ means the time duration in the computational graph $\mathcal{G}(e)$, $\mathbb{1}(i \sim j)$ indicates whether event $i$ is spatially adjacent to event $j$, i.e. they share a vertex. Meanwhile, temporally proximate event pairs are assigned with larger weights of $\cos(|t_i - t_j|/\Delta T)$, where $t_i$ denotes the timestamp of event $i$ (Chen & Ying, 2023).

**Edge recurrence, reciprocity, homophily.** Cohesiveness is a good metric to measure the readability level of an explanation based on temporal network motifs (Paranjape et al., 2017) with several involved events. However, cohesiveness is not defined for single-edge explanations, and it provides low scores for explanations with few adjacent edges. Moreover, there are simple but fundamental mechanisms that rule the evolution of networks that are not captured by cohesiveness. We focus on three well-known, leveraged, and studied mechanisms in graph machine-learning research: edge recurrence (Gastinger et al., 2024), edge

reciprocity, and homophily. *Edge recurrence* is the mechanism by which two nodes that interacted in the past are likely to interact again in the future. In directed networks, *reciprocity* refers to the mechanism where the existence of a directed edge from a node $u$ to node $v$ increases the likelihood of forming a reciprocal edge from $v$ to $u$ in the future. Finally, *homophily* refers to the theory in network science which states that similar nodes may be more likely to attach than dissimilar ones. Since the focus of our work is on the contribution of the edges, we refer to structural homophily, which is the tendency of nodes that share a lot of common neighbors to create a link. Homophily is widely recognized as an interpretable and human-intuitive phenomenon in sociology (McPherson et al., 2001). Similarly, recurrence and reciprocity reflect simple and familiar social behaviors that humans easily recognize and expect, as discussed in the empirical study of evolving social networks (Kossinets & Watts, 2006). Figure 1 shows graphically the limit of cohesiveness on the three mechanisms: reciprocity and edge recurrence cannot be capture by cohesiveness since it required at least two events in $G_{exp}$ to be computed, while explanations based on triadic closure may be penalized by delayed interactions. We introduce three new metrics based on the analog mechanisms to overcome these limitations. In detail, given a target event $e = (u, v, t)$ occurring in $\mathcal{G}(e)$ and its $\mathcal{G}_{exp}(e)$, we define the recurrence as:

$$Recurrence(e) = \begin{cases} w & \text{if } \exists (u, v, t') \in \mathcal{G}_{exp}(e) \text{ s.t. } t' < t \\ 0 & \text{otherwise.} \end{cases}$$

where $w > 0$. For simplicity, we set $w = 1$ for all the events, but it could be the minimum or the maximum among the $w_{t'} = \cos\left(\frac{|t - t'|}{\Delta T}\right)$. Reciprocity can be defined analogously, i.e. it exists at least an event $(v, u, t')$ such that $t' < t$. Finally, we define structural homophily based on the notion of temporal common neighbors:

$$Homophily(e = (u, v, t)) = \frac{|N(u) \cap N(v)|}{\min(|N(u)|, |N(v)|)}$$

where $N(u)$ and $N(v)$ are the sets of temporal neighbors in $\mathcal{G}_{exp}(e)$ of nodes $u$ and $v$.

**What defines the best explainability technique for discrete-time GNNs?** Following previous works (Chen & Ying, 2023), the best technique reaches the highest level of temporal fidelity in the minimum execution time. On average, its explanations are cohesive and leverage recurrence, reciprocity, and homophily, when present in the dataset.

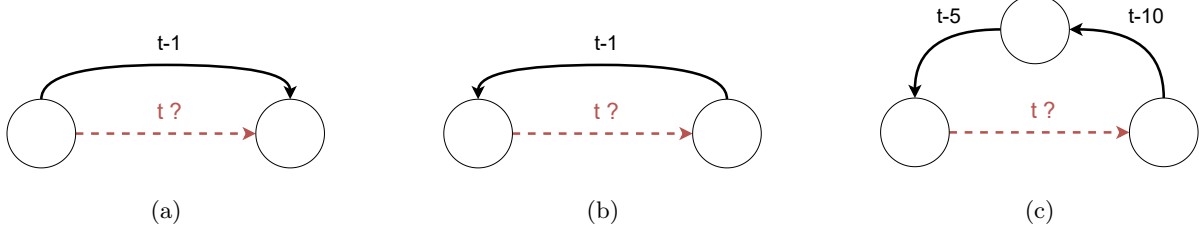

Figure 1: Pitfalls of cohesiveness, which is not defined for edge recurrence (a) and reciprocity (b), and it is penalized by temporal delay in structural homophily (c). However, these 3 explanations are very human intelligible.

**Aspect 3: baselines and models**

**Baselines.** Previous works (Xia et al., 2023; Chen & Ying, 2023; He et al., 2022) do not compare their models against simple baselines tailored for temporal networks, which can make it challenging to gauge the progress made in addressing the temporal aspects of the explanations. To this end, we introduce two simple baselines to create edge masks for temporal graph explainability:

- *LastSnapshot-Explainer*: it sets the edge mask with 1s for the candidate events in the computational graph that happens in the snapshot before the target event and with 0s for all the other events, based on the idea that the most recent events have the great impact on the target one.

- *TemporalNeighbors-Explainer*: given a target event $e = (u, v, t)$, it construct the $2-$hop induced subgraph of $\mathcal{G}(e)$ around nodes $\{u, v\}$, preserving paths with non decreasing timestamps from neighbors to $u$ and $v$. Then, it sets the edge mask with 1s for the events involved in the constructed subgraph and 0s for the rest. It can be seen as a simple, non-parametric method that selects subgraphs by matching events that preserve the temporal graph's causal topology, without relying on gradient information or model internals.

**Discrete-time models.** To the best of our knowledge, there are no methods tailored for discrete-time GNNs in literature. Hence, we extend two well-known approaches for graph explainability to handle discrete-time temporal networks. Specifically, we choose GNNExplainer (Ying et al., 2019) and PGExplainer (Luo et al., 2020) as they are the most representative of non-generative and generative methods, respectively. *GNNExplainer* formalizes the concept of importance using mutual information, leading to the following objective function optimized using gradient descent:

$$\min_{M_E} \quad - \sum_{c=0,1} \mathbb{1}[y = c] \log P_f(Y = y | G = A \odot \sigma(M_E), X = X)$$

where $M_E$ is the edge mask of a static graph, $A$ its adjacency matrix, $X$ its node feature matrix, $c$ the binary class for link prediction. In our setting, given an event $e = (u, v, t)$, $M_E$ is a mask on historical events before $e$, which can be seen as a set of edge masks for each snapshot $M_E = \{M_{E_j}\}_{j=1}^{t-1}$. Hence, the GNNExplainer's objective has been modified to handle discrete-time temporal networks and compute the importance score for each event as follows:

$$\min_{M_E} \quad - \sum_{c=0,1} \mathbb{1}[y = c] \log P_f(Y = y | G = \{A_j \odot \sigma(M_{E_j})\}_{j=1}^{t-1}, X = \{X_j\}_{j=1}^{t-1})$$

where $A_j$ and $X_j$ are the adjacency matrix and node feature matrix of snapshot $j$, respectively. *PGExplainer* adopts a deep neural network to parameterize the generation process of explanations, which enables a natural approach to explaining multiple instances collectively. Specifically, an MLP $h_\theta(j, k)$ infers the importance score of an edge $j$ w.r.t. a target edge $k$. Treating the output of $h_\theta$ as an edge mask assignment weight, PGExplainer can leverage the same objective function of GNNExplainer to update the weights of the MLP on multiple target instances (more details in Luo et al. (2020)). Given a target event $k = (u_k, v_k, t_k)$ and a candidate $j = (u_j, v_j, t_j)$, we can extend the PGExplainer architecture to handle discrete events, constructing the following MLP input features:

$$Z_{k,j} = [H_{u_k} || H_{v_k} || t_k || H_{u_j} || H_{v_j} || t_j || \cos(\frac{|t_k - t_j|}{\Delta T})]^T,$$

where $H_{u_k}$ represents the node embedding of node $u_k$ obtained through the pre-trained TGNN, $\Delta T$ is the computational graph duration, and timestamps are represented as progressive integer numbers.

## 5 Experiments

### 5.1 Experimental setup

**Datasets.** We evaluate the explainability models on three well-known real-world temporal graph datasets, BitcoinOTC (Pareja et al., 2020), Reddit-title (You et al., 2022), and Email-EU (Paranjape et al., 2017), covering the three of the most important applicative domains of discrete-time networks, which are financial, social, and collaboration networks, plus a recent temporal network dataset coming from protein interactions (Fu & He, 2022). BitcoinOTC is a who-trusts-whom network of people who trade using Bitcoin, derived from the homonymous platform. Reddit-title is a hyperlink network representing the directed connections between two communities on Reddit. Email-EU was generated using email data from a large European research institution, where e-mails only represent communication between institution members. Detailed dataset statistics are given in the Supplementary Material.

**Base models.** We adopt three state-of-the-art discrete-time graph neural networks as the base model: EvolveGCN (Pareja et al., 2020), GCRN-GRU (Seo et al., 2018), and ROLAND-GRU (You et al., 2022). EvolveGCN is representative of model evolution methods, while GCNRN-GRU and ROLAND are based on embedding evolution (Longa et al., 2023). Base models are pre-trained using the best configuration of hyperparameters in the original papers, when available, or optimized using grid-search. You can refer to the Supplementary Material for more information and their results for the link prediction task.

**Explainability models.** We compare all the baselines and techniques introduced in this paper (Section 4). We also consider a dummy explainer that outputs random events in the computational graph. Furthermore, we consider Saliency (SA) and Integrated Gradients (IG), which are two of the most effective solutions for static graphs (Amara et al., 2022), but they were never considered for temporal networks (Xia et al., 2023; Chen & Ying, 2023). SA measures edge importance as the weight on every edge after computing the gradient of the output with respect to node embeddings (Baldassarre & Azizpour, 2019), while Integrated Gradient (IG) avoids the saturation problem of Saliency by accumulating gradients over the path from a zero-vector and the input at hand (Sundararajan et al., 2017). Lastly, we include SHAP as one of the most consolidated techniques for explainability and representative of the perturbation-based methods (Lundberg & Lee, 2017).

**Configuration.** We split each dataset chronologically in train, validation, and test sets using 70/10/20 of the snapshots. We randomly sample 50 target events to explain each snapshot. Note that this choice led to test explainability methods on a number of events that is up to five times the number of instances in previous works (Xia et al., 2023; Chen & Ying, 2023), which were 500 overall. For each target, the candidate events are all the edges that appear in a time window immediately before the event, whose size is set equal to the 10% of the snapshots in the dataset. Following previous works (Amara et al., 2022), to compare different explainability techniques, we set the sparsity level of the explanations to a maximum size, equal to 20 events. All the explainers are trained for 200 epochs. Code and Supplementary Material are available in a Github repository[1].

## 5.2 Results

**Temporal fidelity.** We report the temporal fidelities in Table 1. Overall, Integrated Gradients and the discrete-time extension of GNNExplainer reach the highest value of temporal fidelities. In particular, gradient-based methods, such as Saliency and Integrated Gradients, seem to be the best choice for model evolution techniques. In our intuition, assigning edge importance equal to the edge parameter weight after computing the gradient can easily mimic the evolution of the model's parameters through the internalized RNN. Whereas graph-based methods such as GNNExplainer are more suitable for embedding evolution methods since their decisions depend on the structural - i.e. graph - evolution of the node's neighborhoods captured by the embeddings rather than the general dynamic of the system, on which model evolution techniques focus. In general, the simple baselines obtain lower performance than at least one technique for each task. The general low performance of explainability of GCRN-GRU for Reddit-title is probably related to the low performance of the base model in the link prediction task; hence, due to the known performance drop of explainability techniques when applied to incorrect predictions, as highlighted in Amara et al. (2022). Explaining the decisions made on the Email-EU dataset is particularly challenging, likely due to the very high density of events in this dataset. We investigate in detail this challenge in the Supplementary Material.

**Fidelity trends.** Thanks to the setting presented in Section 4, we can show the explainer's fidelity trends over time. Visualizing and comparing the trends is useful to answer the following questions, which cannot be answered using the tabular result only: *a)* Does the performance of an explainer vary along the test snapshots? *b)* Do different techniques follow the same trend? *c)* Does the best technique strictly dominate the others over time? We show the fidelity trends for explaining the decisions of ROLAND-GRU for Reddit-title in Figure 2a, as an interesting example due to the high IG performance. We observe that Integrated Gradients strictly dominate the other explainers. GNNExplainer shows a very similar trend to IG, but its performance varies significantly over time (e.g. up to more than 20% from snapshot 12 to 18). In contrast, the fidelity trend of PGExplainer follows the one of the random explainer, suggesting that its explanations

---

[1]https://github.com/manuel-dileo/dtgnn-explainer

Table 1: Temporal fidelities of seven different explainers over four datasets and three base models. The best result for each model and dataset is in **bold**, the second best is underlined.

| | | BitcoinOTC | Reddit-title | Email-EU | Temporal PPI |
|---|---|---|---|---|---|
| **EvolveGCN** | Random | $72.50 \pm 06.86$ | $40.06 \pm 06.18$ | $18.79 \pm 05.90$ | $50.50 \pm 19.51$ |
| | LastSnapshot | $67.71 \pm 06.18$ | $26.53 \pm 04.83$ | $18.46 \pm 05.94$ | $25.75 \pm 03.93$ |
| | TemporalNeighbors | $69.57 \pm 07.49$ | $78.53 \pm 5.83$ | $20.71 \pm 06.47$ | $65.25 \pm 26.48$ |
| | SA | $76.50 \pm 06.43$ | $80.18 \pm 04.91$ | **$36.37 \pm 11.12$** | $76.75 \pm 07.14$ |
| | IG | **$76.57 \pm 06.37$** | $89.24 \pm 03.82$ | $26.15 \pm 08.17$ | **$89.50 \pm 13.44$** |
| | GNNExplainer | $73.50 \pm 06.82$ | $58.82 \pm 06.40$ | $27.73 \pm 09.37$ | $81.50 \pm 14.20$ |
| | PGExplainer | $70.57 \pm 06.97$ | $40.06 \pm 06.18$ | $18.79 \pm 05.90$ | $50.50 \pm 19.51$ |
| | SHAP | $70.00 \pm 17.32$ | **$90.00 \pm 13.42$** | $35.67 \pm 18.87$ | $82.95 \pm 17.11$ |
| **GCRN-GRU** | Random | $46.86 \pm 07.90$ | $89.00 \pm 05.23$ | $28.58 \pm 07.54$ | $75.00 \pm 14.90$ |
| | LastSnapshot | $22.93 \pm 07.06$ | $75.24 \pm 06.98$ | $27.79 \pm 10.02$ | $44.75 \pm 11.49$ |
| | TemporalNeighbors | $60.00 \pm 06.70$ | $54.65 \pm 05.92$ | $38.52 \pm 12.20$ | $56.88 \pm 12.12$ |
| | SA | $54.50 \pm 08.10$ | $80.35 \pm 06.03$ | $29.65 \pm 07.59$ | $72.00 \pm 17.23$ |
| | IG | $58.00 \pm 04.96$ | $80.56 \pm 05.99$ | $29.83 \pm 07.56$ | $87.75 \pm 08.03$ |
| | GNNExplainer | **$60.29 \pm 08.58$** | **$89.24 \pm 05.36$** | **$60.77 \pm 08.89$** | **$91.00 \pm 09.06$** |
| | PGExplainer | $45.64 \pm 06.44$ | $88.82 \pm 05.34$ | $27.35 \pm 07.36$ | $74.75 \pm 14.86$ |
| | SHAP | $58.86 \pm 15.32$ | $80.00 \pm 17.89$ | $37.34 \pm 12.65$ | $86.12 \pm 18.03$ |
| **ROLAND-GRU** | Random | $53.64 \pm 06.42$ | $64.41 \pm 07.40$ | $38.54 \pm 08.29$ | $64.75 \pm 13.30$ |
| | LastSnapshot | $43.57 \pm 08.06$ | $57.29 \pm 07.58$ | $28.90 \pm 07.27$ | $22.50 \pm 06.91$ |
| | TemporalNeighbors | $14.07 \pm 04.42$ | $81.47 \pm 06.20$ | $38.56 \pm 08.70$ | $60.75 \pm 10.24$ |
| | SA | $56.79 \pm 07.83$ | $84.18 \pm 04.76$ | $48.73 \pm 10.71$ | $79.00 \pm 09.64$ |
| | IG | $57.29 \pm 06.15$ | **$94.88 \pm 02.92$** | $49.40 \pm 11.50$ | **$96.75 \pm 05.19$** |
| | GNNExplainer | **$88.50 \pm 04.77$** | $85.18 \pm 05.27$ | **$49.41 \pm 11.75$** | $88.75 \pm 06.85$ |
| | PGExplainer | $70.00 \pm 06.82$ | $71.24 \pm 06.89$ | $40.94 \pm 08.85$ | $60.75 \pm 15.20$ |
| | SHAP | $60.71 \pm 21.03$ | $84.00 \pm 16.00$ | $41.23 \pm 12.65$ | $81.46 \pm 16.35$ |

could be unreliable (further details in the Supplementary Material). Finally, the trend of LastSnapshot shows that the models need to look up more than the most recent events to explain the decisions. We report all the fidelity trends of our experiments in the Supplementary Material.

**Computation time.**   We report the performance-computation time plot (Amara et al., 2022) of the considered explainers in Figure 2b. Overall, GNNExplainer obtains the best performances, but Saliency and Integrated Gradients are far more efficient with a drop of about 5% in performance only. A similar result was previously observed in explaining static GNNs (Amara et al., 2022). SHAP shows a very high computational time compared to its competitor, making it difficult to utilize it in practical scenarios. On average, it takes about 25 times the time to compute the explanation for a single target event compared to its competitors. Thus, it is excluded from further analysis.

**Human-intelligible explanations.**   In Section 4 we defined different metrics for quantifying the readability of the explanations given by the models, based on temporal aspects. In Figure 3 we report the scatter plot with temporal fidelity for the four metrics, where edge recurrence and reciprocity are computed only for existing future events. In general, the best techniques are the ones in the upper right corner of the plots. We favor the methods that exhibit high cohesiveness and do not show a drastic drop for the other three metrics. The figure shows that the explainability models generally return highly cohesive explanations for discrete-time GNNs, an opposite trend compared to continuous-time GNNs, where the techniques fail to generate human-intelligible explanations (Chen & Ying, 2023). Specifically, PGExplainer provides the most cohesive explanations, but it exhibits a sharp drop in performance for the other metrics. In contrast, GNNExplainer, SA, and IG output cohesive explanations and are also quite able to capture edge recurrence and reciprocity. Overall, no explainability techniques provide explanations with high homophily. Given the

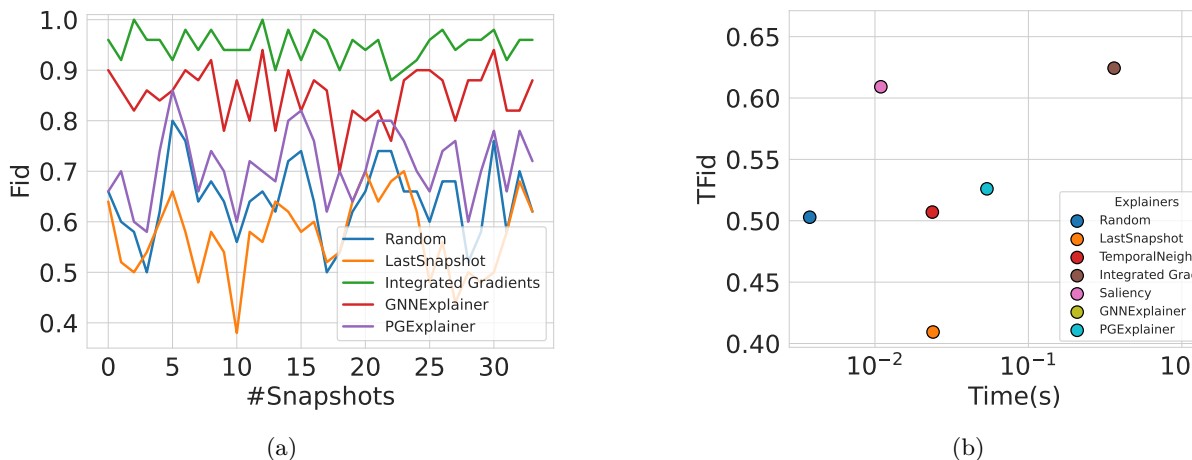

(a)                                                                (b)

Figure 2: (a) Fidelity trends of explainers for the decision made by ROLAND-GRU on future link prediction of Reddit-title dataset. (b) Average temporal fidelity ($y$-axis) and execution time ($x$-axis) for each explainer.

low levels of structural homophily available in the training set of the considered datasets (vertical line in Figure 3d), we considered it not a useful mechanism for the GNNs' decisions.

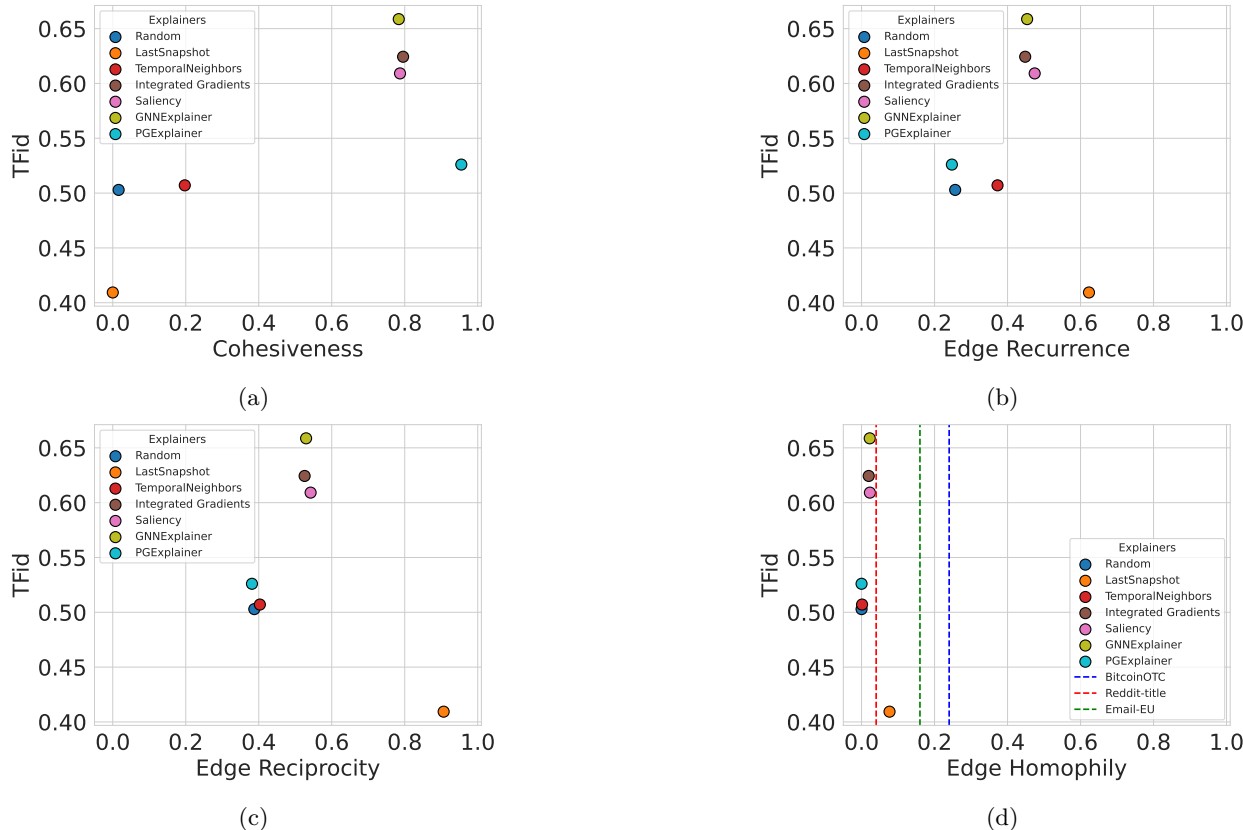

(a)                                                                (b)

(c)                                                                (d)

Figure 3: Performance-readability plots for cohesiveness (a), edge recurrence (b), reciprocity (c), and structural homophily (d). The dashed vertical line in (d) indicates the level of structural homophily in the training set of the selected datasets.

**Case-study: explaining distrust in BitcoinOTC.** Lastly, we focus on explaining the decisions of TGNNs in a specific real-world challenge. Specifically, we chose to explain the decisions behind the existence

of total distrust edge in BitcoinOTC. Predicting the existence of such edges is important because it enables the identification of untrustworthy users and protects the integrity of the platform. Without clear explanations, users and administrators may struggle to justify the model's decisions, particularly when false positives occur. To this end, we ask for an explainability model to obtain the important events related to the decisions of all the distrust edges in the first test snapshot. Overall, we obtain 70 target events. Specifically, we chose ROLAND-GRU as the base model and GNNExplainer, since they achieve the best performance on link prediction and fidelity on BitcoinOTC, respectively. We report three of the most frequent kinds of explanations in Figure 6. We observe highly human-readable explanations. We find that most decisions are made based on edge recurrence, negative consensus on the target nodes, and authority of source nodes. Given a target event $e = (u, v, t)$, we define consensus as the average vote on the incoming edges of the destination node $v$ before $t$, and authority as the in-degree centrality of the source node $u$ before $t$, considering incoming edges with positive votes only. A negative consensus is an average consensus lower than zero. To evaluate quantitatively the presence of these patterns in the given explanations, we compare the distribution of authority and consensus in the explanatory subgraphs, computational graphs, and random vote networks. Random networks are generated using the Erdős–Rényi model (Barabasi & Posfai, 2016) with the probability of edge creation and number of nodes equal to the explanatory graph's density and number of nodes, respectively, and edge weights assigned uniformly at random in $[-10, 10]$. We show the boxplot of the distributions of consensus and authority on the three graphs in Figure 4. Overall, we observe that only a few explanations leverage negative consensus, but they exhibit a value lower than the average consensus of both candidate and random graphs. Concerning authority, we notice that its value is generally far higher in the explanations than in the candidate or random graphs, confirming that it is a very leveraged pattern to decide whether a total distrust edge exists or not.

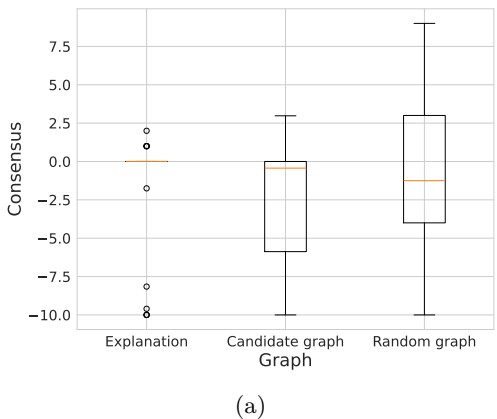

(a)

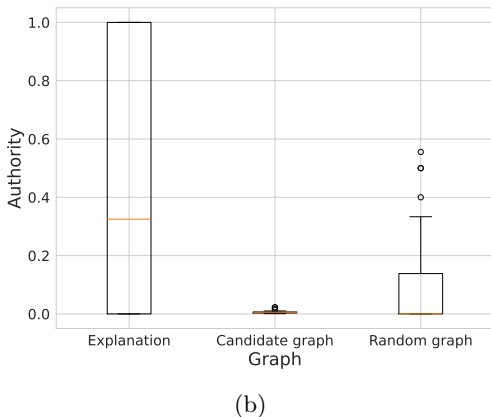

(b)

Figure 4: Boxplot of the distributions of consensus (a) and authority (b) for existing target events corresponding to distrust edges in the first test snapshot. The three boxplots refer to the explanatory subgraphs (Explanation), the computational graph (Candidate graph), and the random graphs.

## 6 Conclusion

In this work, we designed a framework for evaluating explainability techniques on discrete-time graph neural networks, a challenging, unexplored, and important area, since the numerous applications of temporal GNNs in financial, collaboration, and social network analysis. We can outline the best explainability techniques for discrete-time GNNs based on the experiments. Overall, GNNExplainer, Saliency, and Integrated Gradients represent the best choices regarding fidelity, efficiency, and readability trade-off. In terms of performance, the latter two are more tailored for model evolution methods, while GNNExplainer is more suited for embedding evolution models. Saliency and Integrated Gradients are far more efficient than GNNExplainer, while the three techniques exhibit approximately the same level of readability. We summarize the best choices using a decision tree in Figure 5. We hope our framework will push the researchers to develop new techniques in this area.

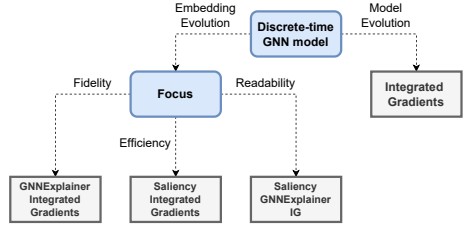

Figure 5: Decision tree for selecting the best explainability techniques for discrete-time GNNs.

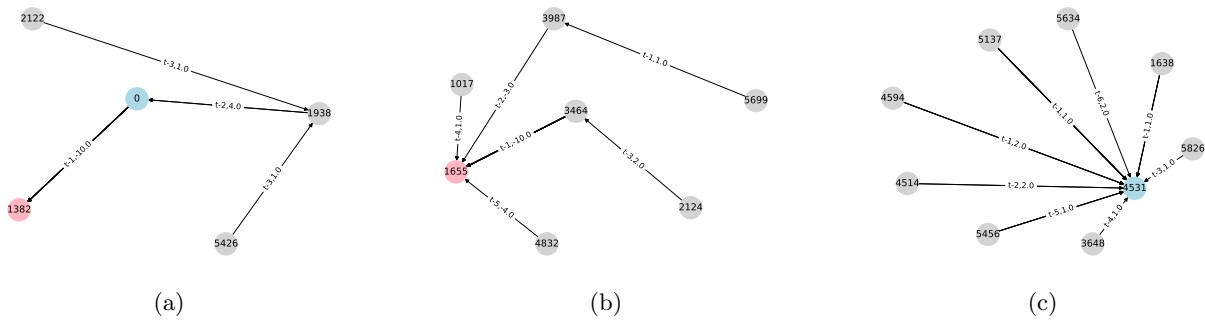

Figure 6: Explanations provided by GNNExplainer for ROLAND-GRU on the existence of total distrust votes (edges) in BitcoinOTC. The node IDs refer to real nodes available in the dataset. Given the target event $e = (u, v, t)$, the light blue node is $u$ and the light pink is $v$. Edges exhibit two labels: the first is the timestamp, and the second is the vote.

## Broader Impact Statement

The explainability of discrete-time temporal graph neural networks has significant implications across domains such as finance, social, or collaboration networks, where decisions must be interpretable and trustworthy. By providing a framework to evaluate explainability techniques, our work contributes positively by enhancing transparency and accountability in AI-driven decision-making. However, potential risks include the misuse of explanations to manipulate or game models, as well as the over-reliance on explainability techniques that may not fully capture the complexity of temporal graph data. System administrators should use explanations as supportive insights rather than definitive answers, ensuring that human expertise remains central to critical decision-making. To mitigate these risks, we encourage the responsible deployment of explainability methods, continuous validation of explanations in real-world settings, and interdisciplinary collaboration to ensure fairness and robustness in applications of temporal GNNs. For instance, one may act at the link prediction level by adopting different methods to increase fairness, as extensively surveyed in Wang et al. (2023).

## Acknowledgments

This work has been partially funded by the Italian Ministry of University and Research (MUR) and the European Union – NextGenerationEU in the framework of the PRIN 2022 project "AWESOME: Analysis framework for WEb3 SOcial MEdia" – CUP: I53D23003680006; by the National Center for Gene Therapy and Drugs Based on RNA Technology—MUR (Project no. CN 00000041) funded by NextGeneration EU program; and by project SERICS (PE00000014) under the NRRP MUR program funded by the EU - NGEU. Views and opinions expressed are however those of the authors only and do not necessarily reflect those of the European Union or the Italian MUR. Neither the European Union nor the Italian MUR can be held responsible for them.

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
