# Supplementary Material: Evaluating explainability techniques on discrete-time graph neural networks

**Manuel Dileo**                                                    *manuel.dileo@unimi.it*
*Department of Computer Science*
*University of Milan, Milan, Italy*

**Matteo Zignani**                                                  *matteo.zignani@unimi.it*
*Department of Computer Science*
*University of Milan, Milan, Italy*

**Sabrina Gaito**                                                   *sabrina.gaito@unimi.it*
*Department of Computer Science*
*University of Milan, Milan, Italy*

**Reviewed on OpenReview:** *https://openreview.net/forum?id=JzmXoOrfry*

## 1  Additional information on datasets

We evaluate the explainability models on four real-world temporal graph datasets, covering four of the most important applicative domains of discrete-time networks, which are financial, social, collaboration, and biological networks. Specifically, we consider:

- BitcoinOTC (Pareja et al., 2020): This is a who-trusts-whom network of people who trade using Bitcoin on a platform called Bitcoin OTC. Since Bitcoin users are anonymous, there is a need to maintain a record of users' reputations to prevent transactions with fraudulent and risky users. Members of Bitcoin OTC rate other members on a scale of -10 (total distrust) to +10 (total trust) in steps of 1.

- Reddit-title (You et al., 2022): It is a hyperlink network that represents the directed connections between two subreddits (a subreddit is a community on Reddit). The network is extracted from publicly available Reddit data covering 2.5 years, from Jan 2014 to April 2017. Specifically, it is extracted from the posts that create hyperlinks from one subreddit to another. A hyperlink originates from a post in the source community and links to a post in the target community. Each hyperlink is annotated with its timestamp. We consider the hyperlinks present in the title of the posts.

- Email-EU (Paranjape et al., 2017): The network was generated using email data from a large European research institution. The e-mails only represent communication between institution members (the core), and the dataset does not contain incoming messages from or outgoing messages to the rest of the world. A directed edge $(u, v, t)$ means that person $u$ sent an e-mail to person $v$ at time $t$. A separate edge is created for each recipient of the e-mail. We consider the subnetwork corresponding to the communication between members of a single department at the institution (Department 1, as defined by SNAP[1]).

- Temporal PPI (Fu & He, 2022): a temporal protein-protein interaction (PPI) network constructed from experimental data measuring dynamic molecular interactions in human cells. Specifically, we consider the KROGAN instance, which includes pairwise interactions between proteins over time as determined by affinity purification mass spectrometry (AP-MS). Each edge (u, v, t) indicates an observed interaction between protein u and protein v at time t. Interactions are timestamped

---

[1]https://snap.stanford.edu/data/email-Eu-core-temporal.html, January 2025

according to the experimental batch in which they were recorded. This dataset is one of the first to provide dynamic PPI networks annotated with temporal information, enabling the modeling of protein complex formation and cellular processes as temporal graphs.

A summary of the dataset properties is given in Table 1. For each dataset, in Table 2 we also report the global level of edge recurrence, as defined in Poursafaei et al. (2022); Gastinger et al. (2024), edge reciprocity, analogously, and structural homophily, defined as the average structural homophily of the edges in the last train snapshot respect to the rest of the train set. Notice that the first three networks are directed but Reddit-title and Email-EU exhibit no reciprocity mechanism in their evolution, and the level of structural homophily is generally quite low.

## 2 Additional information on base models

We adopt three state-of-the-art discrete-time graph neural networks as the base model:

- EvolveGCN (Pareja et al., 2020): it captures the dynamic of the graph sequence of snapshots by using an RNN to update the weights of each GNN layer. In this way, the RNN regulates the GCN model parameter directly and effectively performs model adaptation. Note that the GNN parameters are not trained and only computed from the RNN.

- GCRN-GRU (Seo et al., 2018): It is a generalization of the T-GCN model Zhao et al. (2020), which internalizes a GNN into the GRU cell by replacing linear transformations in GRU with graph convolution operators. GCRN uses ChebNet (Defferrard et al., 2016) for spatial information and separated GNNs to compute different gates of RNNs.

- ROLAND-GRU (You et al., 2022): ROLAND is a framework that can help researchers re-purpose any static GNN to a dynamic graph learning task; consequently, adapting state-of-the-art designs from static GNNs and significantly lower the barrier to learning from dynamic graphs. Specifically, node embeddings at different GNN layers are viewed as hierarchical node states. To generalize a static GNN to a dynamic setting, you only need to define how to update these hierarchical node states based on newly observed nodes and edges. In this paper, we focus on the most effective node update solution, which is based on leveraging gated recurrent units (GRUs).

## 3 Link prediction experiments

**Setting.** We train and evaluate each base model on future link prediction tasks on the three considered datasets. We use our newly introduced training and evaluation setting, which is a slight variation of the recently proposed UTG framework (Huang et al., 2024) in which the training phase is done using the live-update setting (You et al., 2022). As a standard practice (Pareja et al., 2020), we perform random negative sampling for each test snapshot. We report the performance on the test set in terms of the Area Under Precision Recall Curve (AUPRC).

**Hyperparameters.** Base models are pre-trained using the best configuration of hyperparameters in the original papers, when available, or optimized using grid-search. In particular, the best configuration of

Table 1: Dataset statistics

| Dataset | #Edges | #Nodes | Frequency | #Snapshots |
|---|---|---|---|---|
| BitcoinOTC | 35,588 | 6,005 | Weekly | 138 |
| Reddit-title | 571,927 | 54,075 | Weekly | 178 |
| Email-EU | 332,334 | 986 | Daily | 526 |
| Temporal PPI | 14,317 | 3,672 | N/A | 37 |

Table 2: Level of edge recurrence, reciprocity, and structural homophily in the four selected datasets. All metrics range between zero and one.

| Dataset | Recurrence | Reciprocity | Homophily |
|---|---|---|---|
| BitcoinOTC | 0.45 | 0.31 | 0.24 |
| Reddit-title | 0.15 | 0.00 | 0.04 |
| Email-EU | 0.35 | 0.02 | 0.16 |
| Temporal PPI | 0.07 | 0.00 | 0.11 |

Table 3: Best configuration of hyperparameters for Email-EU

| Model | LR | WD | #Layers | $d$ |
|---|---|---|---|---|
| EvolveGCN | 0.010 | 5e-3 | 1 | 128 |
| GCRN-GRU | 0.010 | 5e-3 | 1 | 128 |
| ROLAND-GRU | 0.001 | 5e-3 | 2 | 128 |

hyperparameters can be found in the following work for BitcoinOTC and Reddit-title (Pareja et al., 2020; You et al., 2022). The hyperparameter search spaces used for Email-EU and Temporal PPI are as follows: learning rate (LR) {0.1, 0.01, 0.001, 0.0001}, L2 weight-decay (WD) {5e-1, 5e-2, 5e-3, 5e-4}, number of hidden layers (#Layers) {1, 2}, representation dimension ($d$) {32, 64, 128, 256}. We report the best configuration of hyperparameters for the Email-EU and Temporal PPI datasets in Table 3 and 4, respectively.

**Results.** We report the test-set performance of the models in Table 5. Results are aggregated over the different test snapshots and averaged on 5 different random seeds. Overall, we reach a good level of link prediction performance with at least two models on all the considered datasets. However, GCRN-GRU can obtain performances close to a random edge predictor for Reddit-title and Email-EU datasets. Since the focus of our work is explaining the model's predictions, i.e. the TGNN logic, the level of performance does not impact the discussion of our results. Nevertheless, it could be interesting for future works to analyze if explaining wrong predictions only changes a lot the rank of the evaluated explainability techniques for discrete-time GNNs.

## 4 Fidelity trends

We report all the fidelity trends for each base model and dataset in Figure 5. Thanks to the unlimited space of the Supplementary Material, we decided to report the trends using a subplot for each trend, for better clarity and readability. The figure reported in the paper is Figure h. The same questions discussed in the paper can be answered for each figure.

## 5 Additional discussion on the results

**Additional analysis on PGExplainer fidelity trend.** Reporting the fidelity trends for ROLAND-GRU on Reddit-title, we have shown that the fidelity trend of PGExplainer follows that of the random explainer, suggesting that its explanations could be unreliable. We conducted a deeper empirical investigation to clarify whether this similarity reflects overlapping reliability for the two methods. We reran the evaluation procedure for PGExplainer and the random baseline over 50 independent runs on the Reddit-title dataset, recording the fidelity score at each test snapshot and computing the corresponding confidence intervals. This setup enables us to analyze not just the average behavior of each method but also their variability and potential overlap in performance. The results, presented in Figure 1, show that while PGExplainer and the random baseline follow similar temporal trends, PGExplainer consistently achieves higher average fidelity scores with narrower confidence intervals. Most importantly, the confidence intervals between the two methods exhibit minimal overlap across the test set, indicating that the distributions of fidelity scores

Table 4: Best configuration of hyperparameters for Temporal PPI

| Model | LR | WD | #Layers | $d$ |
|---|---|---|---|---|
| EvolveGCN | 0.010 | 5e-3 | 1 | 128 |
| GCRN-GRU | 0.010 | 5e-3 | 1 | 128 |
| ROLAND-GRU | 0.010 | 5e-3 | 2 | 128 |

Table 5: Link prediction performances of base models on the four datasets, in terms of AUPRC.

| Model | BitcoinOTC | Reddit-title | Email-EU | Temporal PPI |
|---|---|---|---|---|
| EvolveGCN | $84.48 \pm 2.31$ | $\mathbf{88.93 \pm 0.69}$ | $66.91 \pm 6.70$ | $63.72 \pm 11.34$ |
| GCRN-GRU | $96.31 \pm 01.56$ | $53.91 \pm 1.43$ | $57.79 \pm 06.13$ | $\mathbf{66.97 \pm 11.06}$ |
| ROLAND-GRU | $\mathbf{96.89 \pm 1.19}$ | $77.93 \pm 4.19$ | $\mathbf{70.56 \pm 7.14}$ | $54.81 \pm 05.07$ |

are statistically distinguishable despite visual similarities in trend shape. This suggests that the explanations provided by PGExplainer, while structurally similar in evolution to those of a random explainer, are more reliable.

**Fidelity-sparsity curves.** To assess the sensitivity of the evaluated explainability techniques to the sparsity constraint, we computed fidelity–sparsity curves across all three datasets, varying the maximum number of events (edges) allowed in the explanatory subgraphs across the set $\{5, 10, 20, 30, 40, 505, 10, 20, 30, 40, 50\}$, using ROLAND-GRU as the base model. In the main manuscript, a fixed threshold of 20 was applied for all methods, in accordance with previous studies on GNN explainability (Amara et al., 2022). We limited the analysis to a maximum of 50 events, as we believe explanations beyond this size tend to lose interpretability and thus practical utility. This analysis highlights the impact of the sparsity threshold on the quality of explanations in terms of fidelity. Results in Figure 2 show that increasing the threshold generally leads to improvements in fidelity, although the degree of improvement varies across techniques and datasets. Importantly, the relative ranking among methods remains mostly stable, indicating that the comparative performance of techniques is robust to changes in sparsity. This reinforces the utility of using a fixed threshold for fair benchmarking while also underlining the importance of tuning sparsity in real-world deployments where interpretability and performance trade-offs must be balanced. Furthermore, in addition to the observation that the Email-EU dataset presents particular challenges due to its high event density, we notice that all techniques exhibit improved fidelity scores as the sparsity threshold increases. This effect is especially pronounced for PGExplainer and GNNExplainer, i.e., the two methods tailored to discrete-time GNNs, with GNNExplainer achieving over a 20% gain in fidelity and outperforming other approaches at higher thresholds. These findings suggest that the high event density in Email-EU imposes additional constraints on techniques when forced to generate highly compact explanations. Relaxing this constraint helps mitigate the issue, especially for methods designed for discrete-time snapshots.

# 6 Implementation details

Implementing the evaluation framework required extending standard discrete-time GNN pipelines in standard libraries such as Pytorch Geometric. Below, we summarize the main engineering challenges of our solutions:

- Live-update training and inference: Existing libraries such as PyTorch Geometric, PyTorch Geometric Temporal, and Torch-Spatiotemporal assume the deployed setting, where node embeddings are frozen after training. To support live updates, we implemented a custom snapshot-by-snapshot training and inference loop. After each snapshot, the node embeddings are updated based on newly observed edges while keeping the model weights fixed during inference. Future information is never leaked into the model at any step. TGNNs are allowed to be fine-tuned on newly observed events, without requiring the re-training over each snapshot.

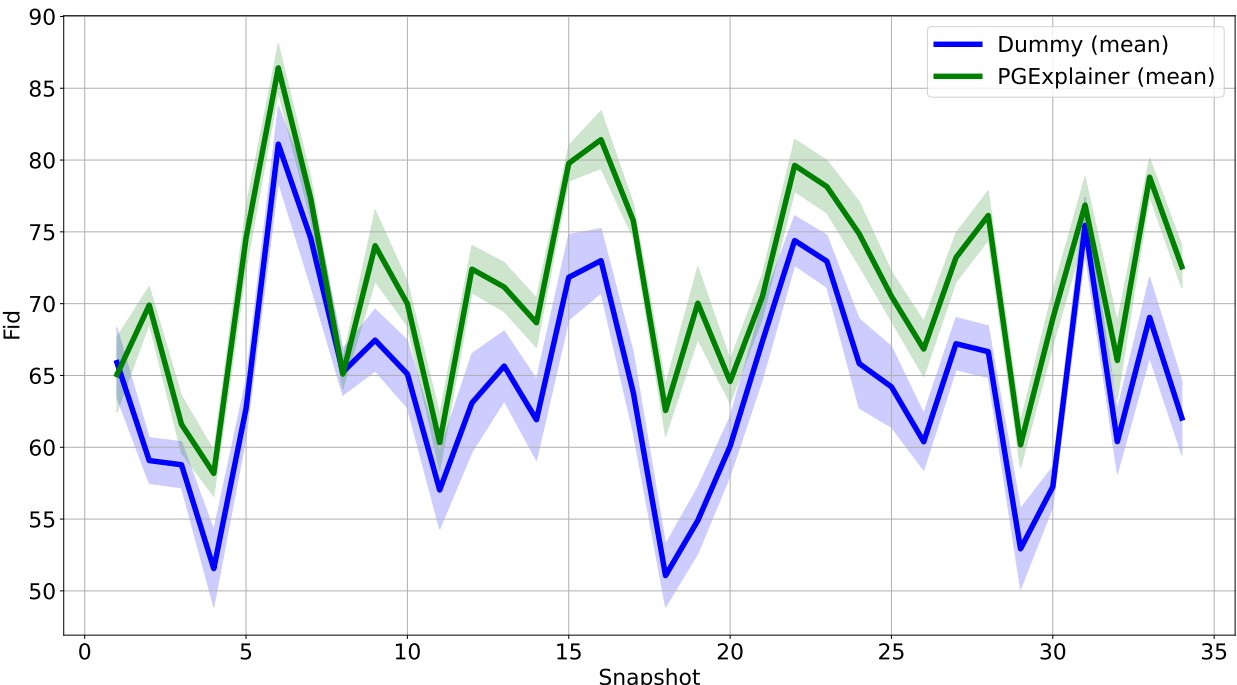

Figure 1: Comparison of fidelity trends between PGExplainer and Random (dummy) explainer on ROLAND-GRU for Reddit-title over 10 independent runs.

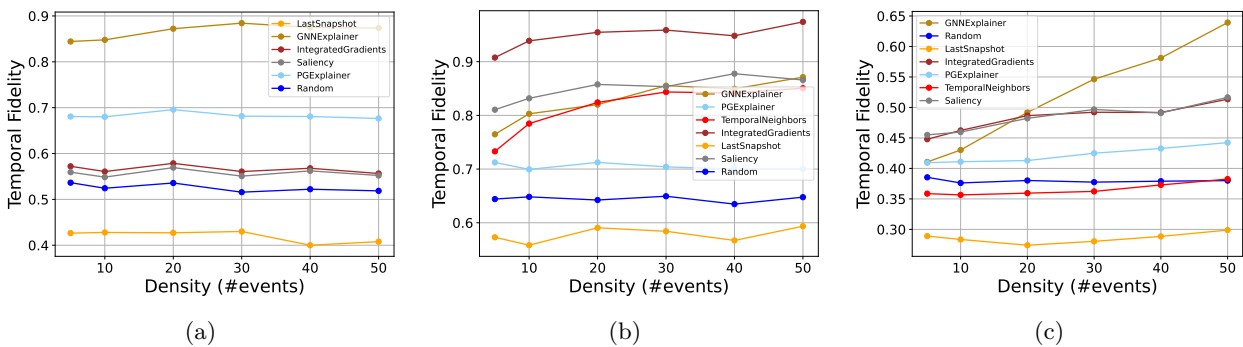

Figure 2: Fidelity-sparsity curves for explainability techniques applied on ROLAND-GRU for Bitcoin-OTC (a), Reddit-title(b), and EmailEU (c). Sparsity is reported as the maximum number of events allowed in the explanatory subgraphs.

- Embedding dictionary: During inference, node embeddings evolve over time. We maintained an explicit dictionary mapping node IDs to their current embeddings. After observing each new snapshot, we updated this dictionary by propagating messages through the latest graph structure. We ensured that only the embedding memory, the updated model, and the current snapshot were kept in RAM to optimize memory usage, avoiding loading the full dynamic graph at once.

- Snapshot-based event sampling: For evaluation, target events had to be sampled at each snapshot individually, rather than uniformly across the entire test set. We extended the data loader to perform per-snapshot sampling, ensuring that explanations are based only on information available up to $t - 1$. This required additional synchronization between the embedding updates and event selection routines.

- Extension of training, inference, and evaluation pipelines: Due to the lack of native support for live-update in existing frameworks, we introduced several custom modules to handle snapshot iteration, embedding caching and updating, event sampling, and synchronization of temporal states. These modifications ensure that the evaluation is fully causal and temporally consistent.

## 7 Additional information on the case study

In the paper, we present a case study where we focus on explaining the decisions of TGNNs in a specific real-world challenge. Specifically, we chose to explain the decisions behind the existence of the total distrust edge in BitcoinOTC. In fact, the BitcoinOTC platform allows its members to rate other members on a scale of -10 (total distrust) to +10 (total trust) in steps of 1. Since their anonymity, this creates a record of users' reputations, which is needed to prevent transactions with fraudulent and risky users. Figure 3 shows the distribution of ratings in BitcoinOTC. Most users receive scores from 1 to 3, and only a few votes are negatives. Hence, predicting the existence of total distrust edges is important because it enables the identification of untrustworthy users, safeguards transactions, and protects the integrity of the platform. However, explaining the decisions made by a GNN in this context is fundamental to fostering trust in the model and ensuring its reliability. Without clear explanations, users and administrators may struggle to justify the model's decisions, particularly when false positives occur. Providing these explanations is crucial to avoid misclassifying good users as fraudulent, which could unfairly damage their reputation and undermine confidence in the system. To this end, we ask for an explainability model to obtain the important events related to the decisions of all the distrust edges in the first test snapshot. Overall, we obtain 70 target events. Specifically, we chose ROLAND-GRU as the base model and GNNExplainer, since they achieve the best performance on link prediction and fidelity on BitcoinOTC, respectively. In the paper, we report three of the most frequent kinds of explanation, observing highly human-readable explanations, and finding that most decisions are made based on edge recurrence, negative consensus on the target nodes, and authority of source nodes. We recall that recurrence is the mechanism by which two nodes that interacted in the past are likely to interact again in the future. Given a target event $e = (u, v, t)$, we define consensus as the average vote on the incoming edges of the destination node $v$ before $t$, and authority as the in-degree centrality of the source node $u$ before $t$, considering incoming edges with positive votes only. A negative consensus is an average consensus lower than zero. To evaluate quantitatively the presence of these patterns in the given explanations, we compare the distribution of authority and consensus in the explanatory subgraphs, computational graphs, and random vote networks. Random networks are generated using the Erdős–Rényi model (Barabasi & Posfai, 2016) with the probability of edge creation and number of nodes equal to the explanatory graph's density and number of nodes, respectively, and edge weights assigned uniformly at random in $[-10, 10]$. Comparing the two metrics with the ones obtained on a random network with an equal number of nodes and edges (on average) is a way to understand whether obtaining the described behavior for consensus and authority only happens by chance or not. We show the boxplot of the distributions of consensus and authority on the three graphs in Figure 4. Overall, we observe that only a few explanations leverage negative consensus, but they exhibit a value lower than the average consensus of both candidate and random graphs. Concerning authority, we notice that its value is generally far higher in the explanations than in the candidate or random graphs, confirming that it is a very leveraged pattern to decide whether a total distrust edge exists or not.

## 8 Extending to node-level tasks

To illustrate the broader applicability of our evaluation framework beyond temporal link prediction, we include an additional experiment on a node classification task. Since the main focus of this work is temporal link prediction—widely recognized as the most studied and adopted task in the TGNN explainability literature—we report these node classification results in the Supplementary Material, serving as a demonstration of our framework's flexibility rather than a comprehensive benchmarking effort.

Specifically, we introduce a temporal node classification task using a spatio-temporal traffic dataset derived from the Montevideo bus network. The dataset contains hourly passenger inflow measurements at the bus stop level for 11 bus lines in Montevideo (Uruguay) during October 2020. Nodes represent bus stops,

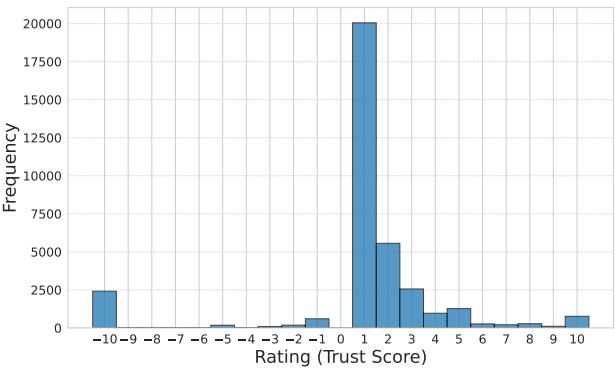

Figure 3: Distribution of ratings in BitcoinOTC.

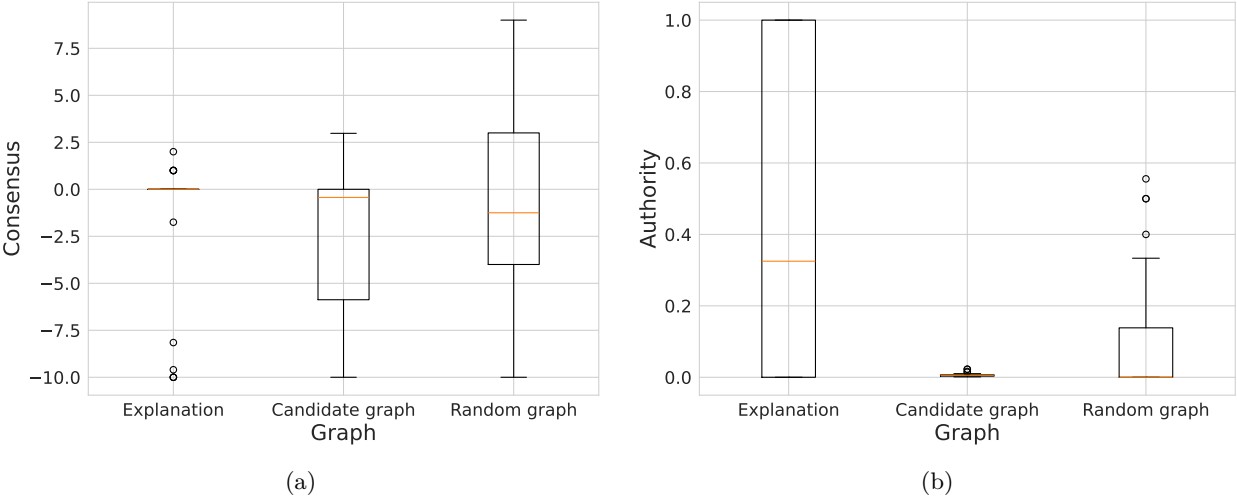

Figure 4: Boxplot of the distributions of consensus (a) and authority (b) for existing target events corresponding to distrust edges in the first test snapshot. The three boxplots refer to the explanatory subgraphs (Explanation), the computational graph (Candidate graph), and the random graphs.

and edges connect stops served consecutively by the same line, with edge weights encoding road distance (Rozemberczki et al., 2021). The goal is to classify whether the inflow at each bus stop will increase, decrease, or remain approximately constant in the next hour. Given the illustrative nature of this experiment, we perform this evaluation using only ROLAND as base model. The temporal fidelity results are reported in Table 6. Results show that most of the techniques outperform the temporal baselines for this task, highlighting the potential of our framework for the explainability of node-level tasks. Notably, the two best-performing techniques are the ones more tailored for graph-structured data.

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

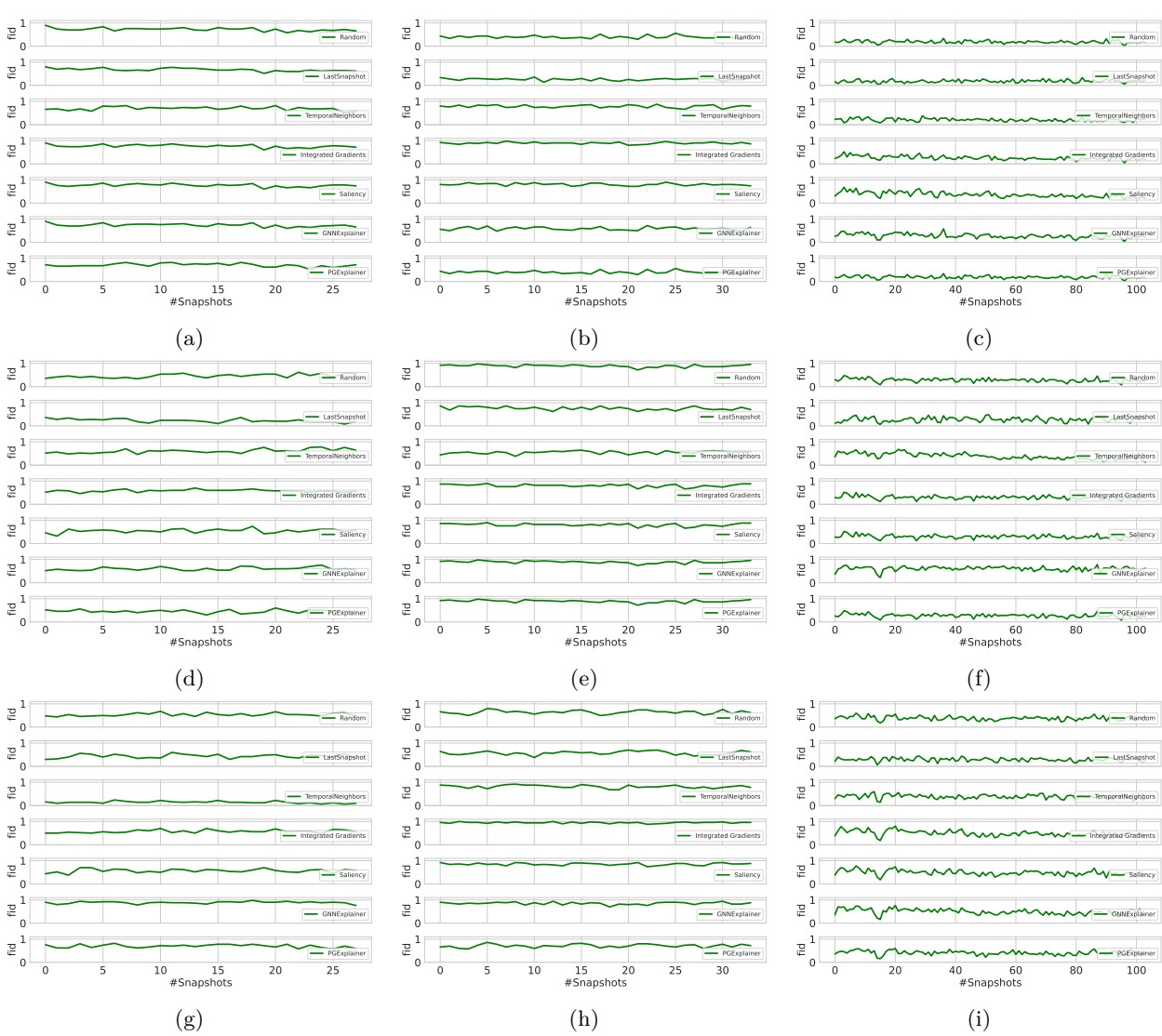

Figure 5: Fidelity trends. The first column refers to the BitcoinOTC dataset, the second to Reddit-title, and the third to Email-EU. The first row refers to EvolveGCN, the second to GCRN-GRU, and the third to ROLAND-GRU.

Table 6: Temporal fidelity for the various explainability techniques on ROLAND-GRU for a temporal node classification task.

| Model | TFid |
|---|---|
| Random | 80.50 ± 07.53 |
| LastSnapshot | 79.78 ± 07.55 |
| TemporalNeighbors | 80.03 ± 07.43 |
| SA | 80.52 ± 07.55 |
| IG | 82.31 ± 07.06 |
| GNNExplainer | **88.30 ± 06.17** |
| PGExplainer | 82.47 ± 09.45 |
| SHAP | 81.32 ± 12.15 |

*formation Processing Systems*, NIPS'16, pp. 3844–3852, Red Hook, NY, USA, 2016. Curran Associates Inc.

Dongqi Fu and Jingrui He. Dppin: A biological repository of dynamic protein-protein interaction network data. In *2022 IEEE International Conference on Big Data (Big Data)*, pp. 5269–5277, 2022. doi: 10. 1109/BigData55660.2022.10020904.

Julia Gastinger, Christian Meilicke, Federico Errica, Timo Sztyler, Anett Schülke, and Heiner Stuckenschmidt. History repeats itself: A baseline for temporal knowledge graph forecasting. In *IJCAI*, pp. 4016–4024. ijcai.org, 2024.

Shenyang Huang, Farimah Poursafaei, Reihaneh Rabbany, Guillaume Rabusseau, and Emanuele Rossi. UTG: Towards a unified view of snapshot and event based models for temporal graphs. In *The Third Learning on Graphs Conference*, 2024. URL `https://openreview.net/forum?id=ZKHV6Cpsxg`.

Ashwin Paranjape, Austin R. Benson, and Jure Leskovec. Motifs in temporal networks. In *WSDM*, pp. 601–610. ACM, 2017.

Aldo Pareja, Giacomo Domeniconi, Jie Chen, Tengfei Ma, Toyotaro Suzumura, Hiroki Kanezashi, Tim Kaler, Tao Schardl, and Charles Leiserson. Evolvegcn: Evolving graph convolutional networks for dynamic graphs. In *Proceedings of the AAAI conference on artificial intelligence*, pp. 5363–5370, 2020.

Farimah Poursafaei, Shenyang Huang, Kellin Pelrine, and Reihaneh Rabbany. Towards better evaluation for dynamic link prediction. In *NeurIPS*, 2022.

Benedek Rozemberczki, Paul Scherer, Yixuan He, George Panagopoulos, Alexander Riedel, Maria Astefanoaei, Oliver Kiss, Ferenc Beres, Guzman Lopez, Nicolas Collignon, and Rik Sarkar. PyTorch Geometric Temporal: Spatiotemporal Signal Processing with Neural Machine Learning Models. In *Proceedings of the 30th ACM International Conference on Information and Knowledge Management*, pp. 4564–4573, 2021.

Youngjoo Seo, Michaël Defferrard, Pierre Vandergheynst, and Xavier Bresson. Structured sequence modeling with graph convolutional recurrent networks. In *Neural Information Processing: 25th International Conference, ICONIP 2018, Siem Reap, Cambodia, December 13-16, 2018, Proceedings, Part I 25*, pp. 362–373. Springer, 2018.

Jiaxuan You, Tianyu Du, and Jure Leskovec. Roland: Graph learning framework for dynamic graphs. In *Proceedings of the 28th ACM SIGKDD Conference on Knowledge Discovery and Data Mining*, KDD '22, pp. 2358–2366, New York, NY, USA, 2022. Association for Computing Machinery. ISBN 9781450393850. doi: 10.1145/3534678.3539300. URL `https://doi.org/10.1145/3534678.3539300`.

Ling Zhao, Yujiao Song, Chao Zhang, Yu Liu, Pu Wang, Tao Lin, Min Deng, and Haifeng Li. T-GCN: A temporal graph convolutional network for traffic prediction. *IEEE Transactions on Intelligent Transportation Systems*, 21(9):3848–3858, sep 2020. doi: 10.1109/tits.2019.2935152. URL `https://doi.org/10.1109%2Ftits.2019.2935152`.