# OpenReview forum: "Evaluating explainability techniques on discrete-time graph neural networks"
_TMLR — Accepted by TMLR_

### Review · Reviewer_3BDt · 2025-03-31

**Summary Of Contributions:**

The paper introduces a novel framework for evaluating explainability techniques tailored to discrete-time temporal Graph Neural Networks (TGNNs), which model evolving graph-structured data in applications like social networks, financial systems, and collaboration networks.

**Audience:**

Yes

**Claims And Evidence:**

Yes

**Requested Changes:**

- Expand Explainability Methods

Include more explainability techniques, such as a perturbation-based method (e.g., SHAP) or a surrogate model approach, to broaden the comparison. Please see this Github page to select several state-of-the-art and cutting-edge methods: https://github.com/flyingdoog/awesome-graph-explainability-papers. The algorithms tested in the paper, such as GNNExplainer and PGExplainer, are too old.

- Diversify Datasets and Tasks

Test the framework on an additional dataset (e.g., a traffic or biological network) and consider an alternative task (e.g., node classification).

- Expand theoretical background

 Providing a more thorough theoretical justification for the chosen evaluation metrics would strengthen the contribution and make the methodology more robust. For instance, provide a theoretical or empirical justification for the choice of $w=1$ in Recurrence (Recurrence $(e)=w$ if $\exists\left(u, v, t^{\prime}\right) \in \mathcal{G}_{\exp }(e)$ s.t. $t^{\prime}<t$ ) and similar simplifications. Conduct a small user study or cite prior work to validate the humanreadability of Recurrence, Reciprocity, and Homophily.

**Strengths And Weaknesses:**

**Strength**

- The paper addresses a significant gap in the explainability of discrete-time temporal Graph Neural Networks (TGNNs), an underexplored area despite their widespread use in critical domains like finance and social networks. The proposed framework is a pioneering effort to tailor explainability techniques to the unique challenges of discrete-time TGNNs, distinguishing it from prior work focused on static or continuous-time GNNs.

- Extensive Experimentation: The experiments cover three real-world datasets (BitcoinOTC, Reddit-title, Email-EU) and three state-of-the-art TGNN models (EvolveGCN, GCRN-GRU, ROLAND-GRU), providing a robust empirical basis. The inclusion of a case study on distrust in BitcoinOTC enhances practical relevance.

- Insightful Metrics and Analysis: The introduction of new metrics like edge recurrence, reciprocity, and structural homophily, alongside extensions like Temporal Fidelity (TFid = $\frac{1}{T_{\text {test }}} \sum_{j=1}^{T_{\text {test }}}\left(1-\operatorname{Fid}\left(K_j\right)\right)$ ), addresses temporal dynamics effectively. The performance readability trade-off analysis (e.g., Figure 3) is particularly insightful for practitioners.


**Weakness**

- Limited Scope of Explainability Methods: While the paper extends GNNExplainer and PGExplainer, it omits other potentially relevant methods (e.g., SHAP or LIME adaptations) that could provide a broader comparison. This limits the generalizability of the findings. Additional baselines could further strengthen the claims, especially by including newer explainability techniques.

- Dataset and Model Bias: The choice of datasets and base models, while representative, may not fully capture the diversity of discrete-time TGNN applications (e.g., biological networks or traffic systems). The focus on link prediction as the sole task restricts the framework's applicability.

- Some theoretical justifications for the chosen evaluation metrics are not fully explored, leaving some aspects of the approach underexplained. The new metrics (Recurrence, Reciprocity, Homophily) are innovative but lack a detailed justification for their weighting schemes (e.g., 𝑤=1 in Recurrence). Their correlation with human interpretability is assumed rather than empirically validated.

---

> ### Author Response · Authors · 2025-04-29
> **Expand explainability methods and addition of datasets**
>
> We thank the reviewer for finding that our work addresses a significant gap in the explainability of discrete-time temporal Graph Neural Networks (TGNNs), that we conducted extensive experimentation, and introduced Insightful Metrics and Analysis. Below, we addressed in detail the requested changes. We hope our responses address the concerns raised, and we remain available for any further clarifications. We will submit an edited version of our manuscript incorporating all the changes in the next few days.
>
> **(W1) Expand Explainability Methods**: We included SHAP as a representative of perturbation-based techniques to broaden the comparison, as suggested by the reviewer. This version of SHAP is tailored for discrete-time GNNs, and it perturbs past events instead of node features. We report the temporal fidelity results of SHAP in the following table:
>
> |           | BITCOINOTC    | REDDIT-TITLE  | EMAIL-EU      |
> | --------- | ------------- | ------------- | ------------- |
> | EvolveGCN | 70.00 ± 17.32 | 90.00 ± 13.42 | 35.67 ± 18.87 |
> | GCRN-GRU  | 58.86 ± 15.32 | 80.00 ± 17.89 | 37.34 ± 12.65 |
> | ROLAND    | 60.71 ± 21.03 | 84.00 ± 16.00 | 41.23 ± 12.65 |
>
> In general, SHAP obtains comparable performances with other techniques, outperforming previous tested solutions, especially on EvolveGCN, highlighting that it may be a profitable choice to explain DTGNNs. However, it exhibits higher variability of fidelity for all the experiments. Moreover, concerning execution time, it shows a very high computational time compared to its competitor, making it difficult to utilize it in practical scenarios. On average, it takes 25 times the time to compute the explanation for a single target compared to its competitors (Indeed, we needed all three weeks to compute the results on our A100 GPU).
>
> **(W2) Diversify Datasets**: We thank the reviewer for suggesting two other discrete-time temporal GNN applications. We added a temporal protein-to-protein interaction network collected by [1] as a representative of the biological network realm. We report the temporal fidelities in the following table:
>
> |                   | EvolveGCN     | GCRN-GRU      | ROLAND        |
> | ----------------- | ------------- | ------------- | ------------- |
> | Random            | 50.50 ± 19.51 | 75.00 ± 14.90 | 64.75 ± 13.30 |
> | LastSnapshot      | 25.75 ± 03.93 | 44.75 ± 11.49 | 22.50 ± 6.91  |
> | TemporalNeighbors | 65.25 ± 26.48 | 56.88 ± 12.12 | 60.75 ± 10.24 |
> | SA                | 76.75 ± 07.14 | 72.00 ± 17.23 | 79.00 ± 9.64  |
> | IG                | 89.50 ± 13.44 | 87.75 ± 08.03 | 96.75 ± 5.19  |
> | GNNExplainer      | 81.50 ± 14.20 | 91.00 ± 09.06 | 88.75 ± 6.85  |
> | PGExplainer       | 50.50 ± 19.51 | 74.75 ± 14.86 | 60.75 ± 15.20 |
> | SHAP              | 82.95 ± 17.11 | 86.12 ± 18.03 | 81.46 ± 16.35 |
>
> Results follow similar trends compared to the other datasets. Notably, the temporal baselines are outperformed by far in this context. Regarding the traffic prediction dataset, we note that performing link prediction would be less informative in this context, as the underlying road network, i.e., the connectivity between nodes, remains static over time.
>
> [1] Fu et al. (2022). DPPIN: A Biological Repository of Dynamic Protein-Protein Interaction Network Data. IEEE Conference on Big Data

---

> > ### Author Response · Authors · 2025-04-29
> > **Diversify tasks and expand theoretical background**
> >
> > **(W2b) Diversify Tasks**: We added a temporal node classification example leveraging a spatio-temporal dataset of traffic based on the Montevideo bus network. Specifically, we leveraged a dataset of inflow passengers at the bus stop level from Montevideo city. This dataset comprises hourly inflow passenger data at the bus stop level for 11 bus lines during October 2020 from Montevideo city (Uruguay). Vertices are bus stops, edges are links between bus stops when a bus line connects them, and the weight represents the road distance [2]. The objective is to predict whether the passenger inflow in bus stops will increase, decrease, or remain (about) the same in the next hour. Since the main focus of our work is on temporal link prediction, as done in all the previous work about TGNNs’explainability, given that is the most well-known, studied and leveraged task, this extension of our work is useful to provide just an example of the possibility to leverage our framework even for no edge-level task such as node classification; hence, the experiments have been run for ROLAND only, and will be included in the Supplementary Material. The temporal fidelities are reported in the following table:
> >
> > |                   | TFid          |
> > | ----------------- | ------------- |
> > | Random            | 80.50 ± 07.53 |
> > | LastSnapshot      | 79.78 ± 07.55 |
> > | TemporalNeighbors | 80.03 ± 07.43 |
> > | SA                | 80.52 ± 07.55 |
> > | IG                | 82.31 ± 07.06 |
> > | GNNExplainer      | 88.30 ± 06.17 |
> > | PGExplainer       | 82.47 ± 09.45 |
> > | SHAP              | 81.32 ± 12.15 |
> >
> > Results show that most of the techniques outperform the temporal baselines for this task, highlighting the potential of our framework for the explainability of node-level tasks. Notably, the two best-performing techniques are the ones more tailored for graph-structured data.
> >
> > **(W3) Provide a justification for the choice of weight in human-readability metrics**: In our implementation, we set the weight for recurrence and reciprocity to 1, effectively making the corresponding metrics binary, indicating whether at least one recurrent or reciprocal event was leveraged in explaining a target event. This binary formulation aligns with the primary goal of the proposed metrics: to assess whether an explanation captures the presence of these structural mechanisms at all, independent of their strength. Given that our work introduces these metrics as general-purpose tools to evaluate any explainability technique across any temporal graph dataset, the binary version provides a simple, interpretable foundation for comparison. Nonetheless, the metric can be extended to incorporate graded information by assigning weights that reflect, for example, the temporal proximity of the relevant events. For instance, one such formulation could use $w_{t'} = \cos\left(\frac{|t-t'|}{\Delta T}\right)$, where the highest value among the valid events can be taken as the weight, thereby emphasizing more recent (i.e., temporally closer) interactions. This allows the metric to quantify not only whether recurrence or reciprocity is used, but also how strongly recent patterns are emphasized. In application-specific scenarios, users may opt for a different weighting function to better reflect domain knowledge or desired sensitivity. Additionally, reporting the metrics under multiple weighting schemes could offer further insights into the nature of the explanations.
> >
> > **(W4) Cite prior work to validate the human readability of Recurrence, Reciprocity, and Homophily**: We thank the reviewer for the suggestion. Recurrence, reciprocity, and homophily are not only fundamental mechanisms governing network evolution, but also align closely with human intuition about interactions. Homophily, the tendency for similar nodes to connect, is widely recognized as an interpretable and human-intuitive phenomenon in sociology [3]. Similarly, recurrence (repeated interactions between the same nodes) and reciprocity (the tendency to respond to interactions) reflect simple and familiar social behaviors that humans easily recognize and expect, as discussed in the empirical study of evolving social networks [4]. We will add these references to better motivate the human readability of our metrics.
> >
> > [2] Rozemberczki et al. (2021). PyTorch Geometric Temporal: Spatiotemporal Signal Processing with Neural Machine Learning Models. Proceedings of the 30th ACM International Conference on Information and Knowledge Management.
> >
> > [3] McPherson, M., Smith-Lovin, L., & Cook, J. M. (2001). Birds of a feather: Homophily in social networks. Annual Review of Sociology.
> >
> > [4] Kossinets, G., & Watts, D. J. (2006). Empirical analysis of an evolving social network. Science.

---

> > > ### Comment · Reviewer_3BDt · 2025-04-29
> > >
> > > Thanks for your response. One last question: Is it possible to include a non-parametric GNN explanation method as well, e.g., MatchExplainer?
> > >
> > > Ref: Rethinking explaining graph neural networks via non-parametric subgraph matching. ICML 2024.

---

> > > > ### Author Response · Authors · 2025-04-30
> > > >
> > > > Dear reviewer 3BDt,
> > > >
> > > > Thank you for your follow-up and for pointing to MatchExplainer. Our framework is indeed compatible with non-parametric GNN explanation methods and can be used to evaluate and compare them. For example, the temporal neighbors explainer we propose as a simple baseline is itself a non-parametric method: it selects subgraphs by matching events that preserve the causal topology of the temporal graph, without relying on gradient information or model internals.
> > > >
> > > > We agree that designing dedicated non-parametric explainers for discrete-time GNNs, particularly for link prediction tasks, represents a valuable and promising research direction. However, such a development would involve substantial methodological work that extends beyond the scope of our current study. As a case in point, the cited MatchExplainer is designed specifically for graph-level tasks; adapting it to the link prediction setting, especially within temporal graph learning frameworks, would not be straightforward and would likely require significant modifications to both the method and its evaluation pipeline.
> > > >
> > > > We appreciate the suggestion and believe that extending our benchmark with additional non-parametric explainers could be a fruitful avenue for future work.

---

> > > > > ### Comment · Reviewer_3BDt · 2025-04-30
> > > > >
> > > > > Thanks for your reply. Then just mention and cite it in your paper to help readers understand the relationship between your study and existing non-parametric methods. Overall, I am satisfied with the rebuttal and would recommend accepting.

---

> > > > > > ### Author Response · Authors · 2025-05-01
> > > > > >
> > > > > > Dear reviewer 3BDt,
> > > > > >
> > > > > > Thank you for your positive feedback and for the constructive suggestions. As recommended, we have added a reference to MatchExplainer in the revised version of the manuscript.
> > > > > >
> > > > > > Best regards,
> > > > > >
> > > > > > The Authors

---

### Review · Reviewer_eePV · 2025-04-16

**Summary Of Contributions:**

The paper presents a novel framework specifically designed to assess explainability techniques for discrete-time temporal graph neural networks (TGNNs). The key contributions include:

The authors introduce a live-update (prequential) setting that better captures the evolving dynamics of temporal data. This contrasts with traditional approaches where only training-set information is used for generating explanations. By allowing newly observed events from the test set to inform inference, the framework promises more relevant and informative explanations.

The authors define temporal-specific evaluation metrics: they extend standard fidelity measures by incorporating the temporal dimension. Alongside fidelity sufficiency, the authors propose additional metrics such as cohesiveness, edge recurrence, reciprocity, and structural homophily.

The paper tested simple baselines (e.g., LastSnapshot-explainer and TemporalNeighbors-explainer) and more complex methods (GNNExplainer and PGExplainer) offering insights into the trade-offs between fidelity, efficiency, and readability.

Finaly, they work on three real-world datasets covering financial, social, and collaboration networks, providing a comparative analysis of various explainability methods under different TGNN architectures. The results are distilled into performance trends, computation time plots, and a decision tree that guides the selection of appropriate explainability techniques based on application needs.

**Audience:**

Yes

**Broader Impact Concerns:**

Authors describe correctly concerns regarding the broader impact that its work can have in the "Broader Impact Statement" paragraph

**Claims And Evidence:**

Yes

**Requested Changes:**

- critical: provide a deeper analysis about explainability evaluation methods problems and limitations as expressed in "Strenghts and weakness" section
- Add citations in the introduction to better contextualize the work within existing research.
- solving minor weakness

**Strengths And Weaknesses:**

Strengths :
- The paper explores discrete-time TGNNs, an area in explainable AI that has received attention and addressing a gap in existing research. The proposed framework introduces a different evaluation setting and temporal-specific metrics.

- The study is thorough, combining methodological proposals with practical evaluations on diverse datasets and models, and enhancing comparison through different baselines and methods. Detailed experimental setups, including chronological data splits and performance trend analyses, clearly illustrate the framework's practical impact.

-Presentation and style is clear and understandable. The decision tree provided in the conclusion helps readers understand which explainability technique might be best suited depending on whether the focus is on model evolution or embedding evolution, as well as on efficiency versus fidelity trade-offs.

Weaknesses:

- Although the framework is well defined, the paper could benefit from a deeper discussion of its own limitations. For instance: two critical issues that notably affect the performance of explainability frameworks are mentioned but not sufficiently explored or analyzed: "The high performance of random explanations of GCRN-GRU for Reddit-title is related to the low performance of the base model in the link prediction task," and "Explaining the decisions made on the Email-EU dataset is particularly challenging, likely due to the very high density of events in this dataset." A robust evaluation framework for explainability requires a deeper analysys of such problems.

- The introduction section lacks sufficient citations to relevant literature, limiting the contextualization and grounding of the work within existing research.

- "In contrast,  the fidelity trend of PGExplainer follows the one of the random explainer, suggesting that its explanations
 could be unreliable". i think that also this fact has to be analyzed more in detail in an evaluation explainability setup, since PGExplainer it is a well known method for GNN explainability and understanding why its fidelity trend follows the one of the random explainer could be of relevance

Minor:
- Sec.3: "where V is the of nodes..", maybe "set"?
- Fig.3 is small and difficult to analyze, should be enlarged

---

> ### Author Response · Authors · 2025-04-30
>
> We thank the reviewer for their valuable feedback and for finding that our study is thorough, combining methodological proposals with practical evaluations on diverse datasets and models, and enhancing comparison through different baselines and methods. Below, we address its requested changes in detail. We hope our responses address the concerns raised, and we remain available for any further clarifications. We will submit an edited version of our manuscript incorporating all the changes in the next days.
>
> **(W1) Deeper discussion on the limitations**: we thank the reviewer for highlighting the need for a deeper analysis on some issues that arise from the results comparison of the different techniques. Below, we address a discussion on the sentences reported by the reviewer.
>
> > **"The high performance of random explanations of GCRN-GRU for Reddit-title is related to the low performance of the base model in the link prediction task."**
>
> We acknowledge that this phrasing may have been unclear. What we intended to emphasize is a phenomenon observed both in our work and in prior studies, namely, that explainability techniques tend to exhibit a performance drop when applied to incorrect or low-confidence model predictions. This observation is well highlighted in [1], where the authors show that explanations for mispredicted instances are often of lower quality, a limitation that naturally does not apply to random explanations, which are unaffected by the model's prediction correctness. In our case, explainability techniques were applied even to models that did not achieve optimal predictive performance, for instance, GCRN-GRU on Reddit-title. In such scenarios, the fidelity of explanation methods tends to degrade, which can result in unexpectedly high relative performance from random baselines. To clarify this point and better align with prior literature, we will revise the sentence to:
>
> > The general low performance of explainability of GCRN-GRU for Reddit-title is probably related to the low performance of the base model in the link prediction task; hence, due to the known performance drop of explainability techniques when applied to incorrect predictions, as highlighted in [1].
>
> > **"Explaining the decisions made on the Email-EU dataset is particularly challenging, likely due to the very high density of events in this dataset."**
>
> To further investigate this point, we extended our analysis by computing the fidelity-sparsity trade-off on this dataset. Specifically, we varied the maximum number of edges allowed in the explanatory subgraph across the set [5, 10, 20, 30, 40, 50], using ROLAND-GRU as the base model. In the main paper, we followed prior work by fixing this threshold to 20. We cap the threshold at 50 events, as explanatory subgraphs exceeding this size are unlikely to remain interpretable and risk losing practical utility as explanations. Our hypothesis was that, given the dense nature of Email-EU, higher sparsity thresholds might allow the explainability techniques to better capture relevant interactions. Results confirm this intuition: all techniques benefit from increased sparsity thresholds, with improved fidelity scores. Particularly notable is the performance gain of the techniques tailored to discrete-time GNNs—PGExplainer and GNNExplainer. The latter, in particular, achieves more than a 20% increase in fidelity and surpasses the performance of other methods at higher thresholds. These findings suggest that the high event density in Email-EU imposes additional constraints on techniques when forced to generate highly compact explanations. Relaxing this constraint helps mitigate the issue, especially for methods designed for discrete-time snapshots. We will incorporate this additional analysis into the revised version of our work.
>
> > **"In contrast, the fidelity trend of PGExplainer follows the one of the random explainer, suggesting that its explanations could be unreliable."**
>
> We investigate this aspect by running 50 different times the evaluation of the 2 explainability techniques on the specific case study, reporting their performance trend over time with confidence intervals on the multiple runs. Results show that while PGExplainer and the random baseline exhibit similar trends over time, even on several multiple runs, PGExplainer consistently maintains higher fidelity scores with minimal confidence interval overlap, suggesting that its explanations, while structurally similar in evolution, are more reliable. We will include this discussion in the Supplementary material and a pointer to this investigation in the revised manuscript.
>
> [1] Amara et al. (2022). GraphFramEx: Towards Systematic Evaluation of Explainability Methods for Graph Neural Networks. LOG

---

> > ### Author Response · Authors · 2025-04-30
> >
> > **W2 Add citations in the introduction to better contextualize the work within existing research**: We thank the reviewer for the suggestion. In the revised version, we will add citations in the introduction to better contextualize our work within the landscape of existing research on temporal GNNs and explainability.

---

### Review · Reviewer_jQQH · 2025-04-22

**Summary Of Contributions:**

The paper makes four key contributions to the explainability of discrete-time TGNNs: (1) it introduces a new training and evaluation setting that captures the dynamic nature of temporal data; (2) it extends evaluation metrics to account for temporal characteristics like event recurrence; (3) it proposes new baselines for explaining temporal GNNs, including simple models based on recent events and adaptations of existing methods like GNNExplainer; and (4) it provides practical guidance through extensive empirical evaluations highlighting the trade-offs among fidelity, performance, and readability of explanations.

**Audience:**

Yes

**Claims And Evidence:**

Yes

**Requested Changes:**

See weakness. In addition, I have a question:

What type of dataset is used in Figure 2, and how does the model perform on large-scale graph data?

**Strengths And Weaknesses:**

Strengths: The structure of the paper is well-organized with intuitive explanations and detailed experimental results to justify the proposed approach.

Weakness:

1. Although the paper is generally well-structured, I found it difficult to fully grasp the implementation challenges associated with the proposed evaluation procedure, particularly Algorithm 1. This may be partly due to my limited background in temporal graph neural networks; however, I believe the paper would benefit from more detailed, implementation-oriented explanations of the underlying challenges, rather than relying primarily on references to existing methods or simply listing the steps involved in the live-update setting and candidate event selection.

2. We need a comprehensive analysis of hyperparameter sensitivities through either theoretical analysis or numerical justification.

---

> ### Author Response · Authors · 2025-04-29
>
> We thank the reviewer for their valuable feedback and for finding that the structure of the paper is well-organized with intuitive explanations and detailed experimental results to justify the proposed approach. Below, we address its requested changes in detail. We hope our responses address the concerns raised, and we remain available for any further clarifications. We will submit an edited version of our manuscript incorporating all the changes in the next days.
>
> **(W1) Implementation-oriented explanations of the underlying challenges**: We thank the reviewer for the helpful comment. Beyond the conceptual description, there were several practical implementation challenges we had to address for Algorithm 1:
> - Live-update training and inference are not supported out of the box by PyTorch Geometric or even in more tailored temporal GNN libraries such as PyG temporal or torch-spatiotemporal, where training and inference are performed in the deployed setting. We had to implement a custom loop to update node embeddings snapshot-by-snapshot, keeping track of graph evolution over time without leaking future information during training, and allowing fine-tuning of the same TGNN model over time, without retraining it for each snapshot.
> - Embedding updates at test time required modifying the inference pipeline: instead of freezing embeddings after training, we had to cache and update node states after each observed test snapshot, while keeping the model weights frozen. This involved maintaining an explicit mapping between node IDs and their evolving embeddings, and carefully distinguishing the message-passing graph from the predicted one. Furthermore, for memory efficiency, only the current updated model, the node embedding states, and the current snapshot are stored in RAM, without loading the entire graph.
> - Sampling target events per snapshot meant we could not just sample randomly from the whole test set. We implemented snapshot-based sampling, making sure events only depend on information available up to $t-1$, which required changes in the data loader and evaluation pipeline.
>
> Overall, the lack of native support forced us to extend the training, inference, and evaluation pipelines with several custom modules. We will add a dedicated implementation notes section to the appendix to explain these aspects in more detail.
>
> **(W2) Comprehensive analysis of hyperparameter sensitivities**:  We thank the reviewers for the suggestion. Beyond the number of training epochs and optimizer hyperparameters, which were carefully selected following prior work such as GraphFrameEx and the TGNN explainability techniques cited in the manuscript, the only shared hyperparameter across all methods is the sparsity threshold, i.e., the maximum number of events allowed in an explanatory subgraph. In the submitted version, this threshold was set to 20 for all techniques, in line with conventions established in previous studies. To assess the sensitivity of each method to this sparsity constraint, we compute the fidelity-sparsity trade-off curve across the three datasets, varying the threshold in [5, 10, 20, 30, 40, 50], considering ROLAND-GRU as the base model. This analysis allows us to evaluate how the sparsity level affects the performance of the XAI techniques in terms of fidelity. We cap the threshold at 50 events, as explanatory subgraphs exceeding this size are unlikely to remain interpretable and risk losing practical utility as explanations. Results will be available in the next few days with the revised version of the paper.
>
> **(Q1) Figure 2**: Figure 2a reports the fidelity trend for all the explainability techniques applied to the result of the ROLAND-GRU model for the Reddit-title dataset. The model works particularly well on large graph data, as described in the original paper [1]. Figure 2b shows the average temporal fidelity (y-axis) and execution time (x-axis) for each explainer on all the datasets.
>
> [1] You et al. (2022). ROLAND: Graph Learning Framework for Dynamic Graphs. KDD

---

> > ### Author Response · Authors · 2025-05-01
> >
> > Dear reviewer jQQH,
> >
> > We would like to follow up by letting you know that the revised version of the manuscript is now available, along with the supplementary material that includes the full results of the fidelity-sparsity trade-off analysis.
> >
> > The results show that increasing the sparsity threshold generally leads to improvements in fidelity, although the magnitude of this improvement varies across techniques and datasets. Importantly, the relative ranking of the methods remains largely consistent across different threshold values, suggesting that the comparative performance of the techniques is robust to changes in sparsity. This finding supports the use of a fixed threshold for fair benchmarking, while also highlighting the importance of tuning this parameter in practical settings where one must balance interpretability and explanatory performance.
> >
> > We appreciate your feedback, which helped strengthen this aspect of our evaluation.

---

### Author Response · Authors · 2025-05-01
**Revised Manuscript Uploaded – Thank You for your feedback!**

Dear Reviewers,

We sincerely thank you for the time and effort dedicated to reviewing our work, and for the insightful comments and constructive suggestions that have helped us improve the quality and clarity of our manuscript. We are pleased to inform you that we have uploaded the revised version of the paper along with the updated supplementary material.

In this revision, we have addressed the main points raised during the review process, including the following key changes:

- **Expanded Explainability Methods**: In response to Reviewer **3BDt**’s suggestion, we included SHAP as a representative perturbation-based technique to broaden our comparative analysis of explainability methods.

- **Diversified Datasets**: We added a temporal protein–protein interaction (PPI) dataset to test the explainability techniques to include another key application of temporal networks.

- **Diversified Tasks**: We presented a temporal node classification example using a spatio-temporal dataset from the Montevideo public transportation system. These results are presented in the supplementary material.

- **Improved Contextualization and Justification**: We strengthened the background by adding relevant citations in the introduction and cited prior work to support the human interpretability of the metrics introduced in our framework.

- **Deeper Discussion on Limitations**: Following Reviewer **eePV**’s feedback, we conducted additional empirical investigation, presented in the supplementary material, to explore certain claims in more detail. We revised or nuanced the relevant statements in the manuscript accordingly.

- **Implementation-Oriented Challenges**: As requested by Reviewer **jQQH**, we provided a detailed explanation of the implementation challenges associated with our evaluation procedure and Algorithm 1, which are now included in the supplementary material.

- **Hyperparameter Sensitivity Analysis**:  We conducted a fidelity–sparsity trade-off study to evaluate the sensitivity of explainability performance to sparsity constraints, and to investigate whether this influences the relative ranking of the techniques.

- **Minor Corrections**: We addressed minor issues across the manuscript, including typos and figure formatting, as highlighted by the reviewers.

We hope that these revisions comprehensively address your comments and enhance the contribution of our work. Thank you again for your valuable feedback and for helping us strengthen our study.

Sincerely,

The Authors

---

### Decision · Action_Editor_YjK1 · 2025-05-26

**Recommendation:** Accept as is

**Comment:**

All reviewers recommend accepting the papers and are satisfied with the authors' changes. One reviewer wants to see experimental evaluation on larger datasets and against SOTA methods.

**Audience:**

The findings of the present paper will interest a subset of the graph learning community. Hence, the paper meets the sufficient conditions for acceptance.

**Claims And Evidence:**

All claims are sufficiently supported. All reviewers endorsed the acceptance of the present work and stated that the requested changes were adequately addressed.